# Finding process-behavioural parameterisations of a hydrological model using a multi-step process-based calibration and evaluation scheme

**Moritz M. Heuer, Hadysa Mohajerani, and Markus C. Casper**

Department of Physical Geography, University Trier, Trier, 54296, Germany

**Correspondence:** Moritz M. Heuer (heuer@uni-trier.de)

**Abstract.** Process-behavioural hydrological modelling aims not only at predicting the discharge of an area within a model, but also at understanding and correctly depicting the underlying hydrological processes. Here, we present a new approach for the calibration and evaluation of water balance models, exemplarily applied to the Riverisbach catchment in Rhineland-Palatinate, Germany. For our approach, we used the behavioural model WaSiM. The first calibration step is the adjustment of the evapotranspiration (ETa) parameters based on MODIS evapotranspiration data. This aims at providing correct evapotranspiration behaviour of the model and at closing the water balance at the gauging station. In the second step, geometry and transmissivity of the aquifer are determined using the characteristic delay curve (CDC). The portion of groundwater recharge was calibrated using the delayed flow index (DFI). In the third step, inappropriate pedotransfer functions (PTFs) could be filtered out by comparing dominant runoff process patterns under a synthetic precipitation event with a soil hydrological reference map. Then, the discharge peaks were adjusted based on so-called signature indices. This ensured a correct depiction of high-flow volume in the model. Finally, the overall model performance was determined using signature indices and efficiency measures. The results show a very good model fit with values of 0.87 for the NSE (Nash–Sutcliffe model efficiency coefficient) and 0.89 for the KGE (Kling–Gupta efficiency) in the calibration period, as well as an NSE of 0.78 and a KGE of 0.87 for the validation period. Simultaneously, our calibration approach ensured a correct depiction of the underlying processes (groundwater behaviour, runoff patterns). We were also able to detect the model parameterisations based on the PTFs that showed satisfactory results across all calibration steps. This enables a targeted selection of the most suitable PTFs for determining the soil properties. This means that our calibration approach allows selecting a process-behaviourally faithful one from many possible parameterisation variants.

## 1 Introduction

Traditionally, hydrological models are calibrated mainly on the basis of gauging data, with the aim of accurately predicting discharge. However, the underlying processes like groundwater behaviour or runoff generation processes are often neglected in this approach (Schaake et al., 1996; Xiong and Guo, 1999; Casper et al., 2019; Kheimi and Abdelaziz, 2022). Reasons for this could be that data sets for additional calibration steps are missing, more comprehensive calibration is too time-consuming and computationally intensive, or the correctness of certain underlying model processes is insignificant for the specific research question. Relying solely on statistical evaluations of overall runoff performance may not adequately capture model performance for high and low flow extremes (Westerberg et al., 2011; Althoff and Rodrigues, 2021). This means that although these models are then suitable for predicting runoff, they do not allow investigations of the underlying processes. Additionally, the model could behave in unintended ways when incorporating climate or land use changes (Clark et al., 2016). This emphasises the necessity for physically-based models to be not just theoretically accurate but also empirically validated against the dynamics of natural hydrological systems (Beven, 2002).

Process-behavioural modelling addresses this issue by not only considering the discharge but also the discharge-forming processes during model calibration. This approach necessitates the integration of methodological frameworks that align simulated processes with observed catchment responses (Vansteenkiste et al., 2014). For example, studies by Ferket et al. (2010), Zhang et al. (2011), and Meresa et al. (2023) have employed performance metrics to evaluate subsurface flow components, such as interflow and deep percolation to groundwater, within runoff discharge simulations. Similarly, Casper et al. (2023) enhanced the reproduction of spatial and temporal evapotranspiration (ETa) patterns by applying a MODIS-based calibration approach to vegetation-related ETa parameters. Using the example of soil moisture content, Dangol et al. (2023) were able to show that limited approaches to model calibration led to incorrect process depiction. The inclusion of additional data led to an improved representation of the corresponding process in the model. Similar results were obtained by Stisen et al. (2018), who achieved a more robust model calibration by including spatial variables such as soil moisture, remotely sensed land surface temperature, hydraulic head, and actual evapotranspiration in the calibration process in addition to the discharge. Abbas et al. (2024) were able to show that the incorporation of increased parameter numbers paired with the incorporation of different hydrological processes improves the model result. This shows that the use of different hydrologic processes in model calibration is necessary for the correct depiction of the discharge generating processes.

Groundwater's delayed response to precipitation and its role in baseflow during dry periods are critical for accurate water resource management (Beven and Alcock, 2012). The duration from groundwater recharge to baseflow discharge is influenced by topography, geology, vegetation, land use, and climate (Barthel, 2006; Götzinger et al., 2008). Baseflow-fed streamflow is directly related to groundwater storage and its interaction with streams, which can vary heavily across catchments (Barkwith et al., 2015). This complexity necessitates incorporating groundwater flow into hydrological models to accurately simulate discharge under diverse hydrological conditions (Knisel, 1963; Smakhtin, 2001; McNamara et al., 2011; Barkwith et al., 2015; Stoelzle et al., 2015). The behaviour of the groundwater component in water balance models must therefore be considered when calibrating a model. This makes it necessary to implement a way of evaluating the model's ability to correctly represent groundwater behaviour and its temporal contribution to the overall discharge.

Pedotransfer functions (PTFs) allow the estimation of soil hydraulic properties from widely available soil data like grain size, density, or depth. Simulation outcomes of different PTFs highly differ in runoff components (surface runoff, interflow, and deep percolation) and evapotranspiration (ETa) rates in space and time (Refsgaard, 2001; Stisen et al., 2008; Koch et al., 2016, 2017; Casper et al., 2019; Mohajerani

et al., 2021). Therefore, the correct choice of a PTF for soil parameterisation is crucial. Despite the knowledge about the difference the PTF choice makes, modellers seem to give this too little attention. Often, established PTFs are chosen without evaluating if they are really suitable for the soil parameterisation of the specific catchment's soils. This makes it necessary to develop approaches which allow to evaluate if certain PTFs correctly derive the catchment's soil properties, as these fundamentally influence the discharge generation.

Liu et al. (2022) demonstrated that the incorporation of remote sensing data like ETa data or terrestrial water storage change (TWSC) for hydrologic model calibration can improve the depiction of those processes. It was also shown that combinations of different evaluation criteria increase the model accuracy regarding the underlying processes (Nesru et al., 2020; Nolte et al., 2021; Yáñez-Morroni et al., 2024). Also, the relevance of groundwater parameterisations in hydrological models has already been emphasised several times (Troldborg et al., 2007; Troch et al., 2013). However, the calibration of aquifers in hydrological models in particular has so far received too little attention in multi-variable calibration approaches. This results in the need for a calibration scheme that combines approaches for the calibration of surface processes such as evapotranspiration, runoff generation processes, and overall discharge with approaches for the calibration of groundwater behaviour. This is particularly necessary if the model should be used to investigate the effects of changes in environmental variables, for example under changing land uses or under climate change scenarios (Du et al., 2013; Mendoza et al., 2015; Huang et al., 2020). This also applies if the change in discharge-forming processes itself is to be the subject of research (Efstratiadis and Koutsoyiannis, 2010).

To address the above-mentioned challenges, our research introduces a new approach for the parameterisation and calibration of water balance models. This approach comprises the calibration of evapotranspiration patterns of different land uses based on remote sensing ETa data, ensuring correct ETa patterns and a closed water balance. In addition, the ground water behaviour is assessed by deriving the long term baseflow from the measured discharge of the catchment. This allows for calibration of the groundwater behaviour (storage, recession) as well as the groundwater recharge (deep percolation) within the model. Furthermore, the influence of the soil parameterisation on the spatial pattern of runoff generation is assessed. This ensures a correct depiction of runoff patterns over the catchment area. Lastly, high discharge volume is calibrated by deriving information about the catchment discharge characteristics from the flow duration curve. These different methods are applied to model parameterisations whose soil hydrological properties are determined differently by a variety of pedotransfer functions. Therefore, the suitability of individual PTFs to correctly describe the soil properties of the catchment can be evaluated. By incorporating the calibration and evaluation of these different model

aspects, we aim at reaching a model calibration that correctly simulates the discharge as well as the underlying hydrological processes. This represents an advantage over black-box calibration approaches, where the calibration is not aimed at the correct representation of hydrological processes. It also extends existing multi-variable calibration concepts, which previously did not take different soil parameterisations into account in their calibration and evaluation schemes.

The aim of our study is to investigate whether a multi-variable calibration approach can be used to select a model parameterisation that correctly represents both the simulated runoff and the underlying hydrological processes. We hypothesise that (i) the aquifer calibration can be derived from the measured baseflow; (ii) a model parameter set can be found that leads to correct discharge and process depiction; and (iii) soil parameterisations derived by different PTFs that lead to incorrect process depictions in the model can be reliably detected and filtered out.

## 2 Methodology and material

### 2.1 Study area

The Riverisbach catchment (Fig. 1) was selected as the study area for the demonstration of the parameterisation approach. This was due to the good availability of data on soil, land use, ETa patterns, and discharge, which is necessary for the evaluation of the model calibration. The catchment basin is located south-east of Trier in Rhineland-Palatinate, Germany. It covers an area of around $15.42\,\mathrm{km}^2$ and ranges from 329 m above sea level in the north-west to 705 m above sea level in the south, resulting in a height amplitude of 376 m and an average slope gradient of 4.49 %. The used gauging station "Riveristalsperre" is located in the west of the catchment at 49°41.771′ N, 6°46.741′ E. The mean annual precipitation amounts to $918\,\mathrm{mm\,yr}^{-1}$.

The area is located above bedrock from the Drohntal strata, i.e. quartz sandstone and quartzitic sandstone with intercalations of claystone and siltstone. The soils are dominated by Cambisols while Gleysols and Stagnosols can be found along the watercourses in the floodplain area. The majority of the Riverisbach catchment area is covered by forest. Conifers dominate the north-east and west and deciduous trees dominate in the centre and south. In the west there are also small areas of grassland and mixed woodland.

### 2.2 Data sources

Soil type information was taken from the "Bodenflächendaten im Maßstab 1 : 50 000 (BFD50)" (Landesamt für Geologie und Bergbau, 2021). The data for the land use are derived from Corine land cover (ISPRA – Istituto Superiore per la Protezione e per la Ricerca Ambientale, 2018) as well as from the European Union's Copernicus Land Monitoring Service information (European Environment Agency

(EEA), 2018). INTERMET data (Gerlach, 2006) were used as time series for meteorological data. Wind data were taken from the Agrarmeteorologie Rheinland-Pfalz (2024). Values for the saturated hydraulic conductivity $k_{\mathrm{sat}}$ were taken from Ad-hoc-AG Boden (2005).

### 2.3 Model setup and parameterisation

The WaSiM model (Schulla, 1997) version 10.08.02 (Schulla, 2024a) was selected for the simulation and development of the parameterisation approach. It is a deterministic, hydrological catchment model that is suitable for the simulation of both small ($< 1\,\mathrm{km}^2$) and very large ($> 10\,000\,\mathrm{km}^2$) areas. It also simulates the underlying processes that lead to discharge generation. This includes the ETa, groundwater flow, surface runoff, and interflow, as well as groundwater recharge. It is therefore suitable for a process-behavioural modelling approach that includes the calibration of these processes. A schematic depiction of the WaSiM model is shown in Fig. 2. The soil is represented in the model as a rectangular grid of 1-dimensional columns. Each of these columns is divided into soil horizons of different thicknesses, which in turn are subdivided into several layers. At the bottom, a section of aquifer layers is included. Surface runoff, interflow, and groundwater-contributing deep percolation can be generated. Surface runoff and interflow of each subcatchment are delayed through a single linear reservoir (SLR) each. Snowmelt is handled with a temperature-index-approach where the snowmelt rate is determined by the temperature and a melt factor.

Spatially resolved data are differentiated within the model using grid structures. This also enables the model to interpolate climatic input data over the catchment area. The model uses the Richards equation (Richards, 1931) to calculate the water transport within the unsaturated soil zone. It is defined as:

$$\frac{\partial \theta}{\partial t} = \frac{\partial}{\partial z}\left[k(\Psi_{\mathrm{m}})\left(\frac{\partial \Psi_{\mathrm{m}}}{\partial t}\right)\right], \tag{1}$$

where $z$ is the depth, $\theta$ is the water content [vol %], $t$ is the time [d], and $k(\Psi_{\mathrm{m}})$ is the hydraulic conductivity in dependence of the matrix potential $[\mathrm{cm\,d}^{-1}]$. The van Genuchten parameters (Van Genuchten, 1980) are used to calculate the soil physical properties. The Penman–Monteith (Monteith, 1965) method is used to calculate evapotranspiration. A two-dimensional approach based on Darcy's law (Darcy, 1856) is used to calculate groundwater flow. It is defined as:

$$q = k \cdot \frac{\partial \Psi}{\partial z}, \tag{2}$$

where $q$ is the volume flow $[\mathrm{m}^3\,\mathrm{s}^{-1}]$, $k$ is the hydraulic conductivity $[\mathrm{m\,s}^{-1}]$, and $[\partial \Psi / \partial z]$ is the hydraulic gradient [–].

For the model parameterisation, a spatial resolution of 40 m and a temporal resolution of 1 h were chosen. The

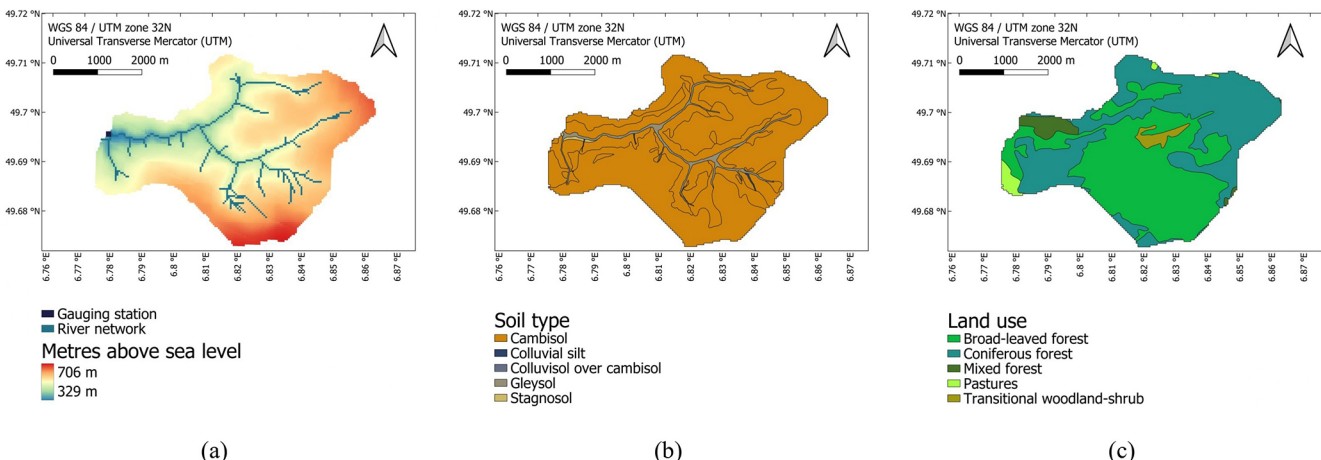

(a)                             (b)                             (c)

**Figure 1.** Topography, soil types and land cover types within the Riverisbach catchment as it's used within our WaSiM based model.

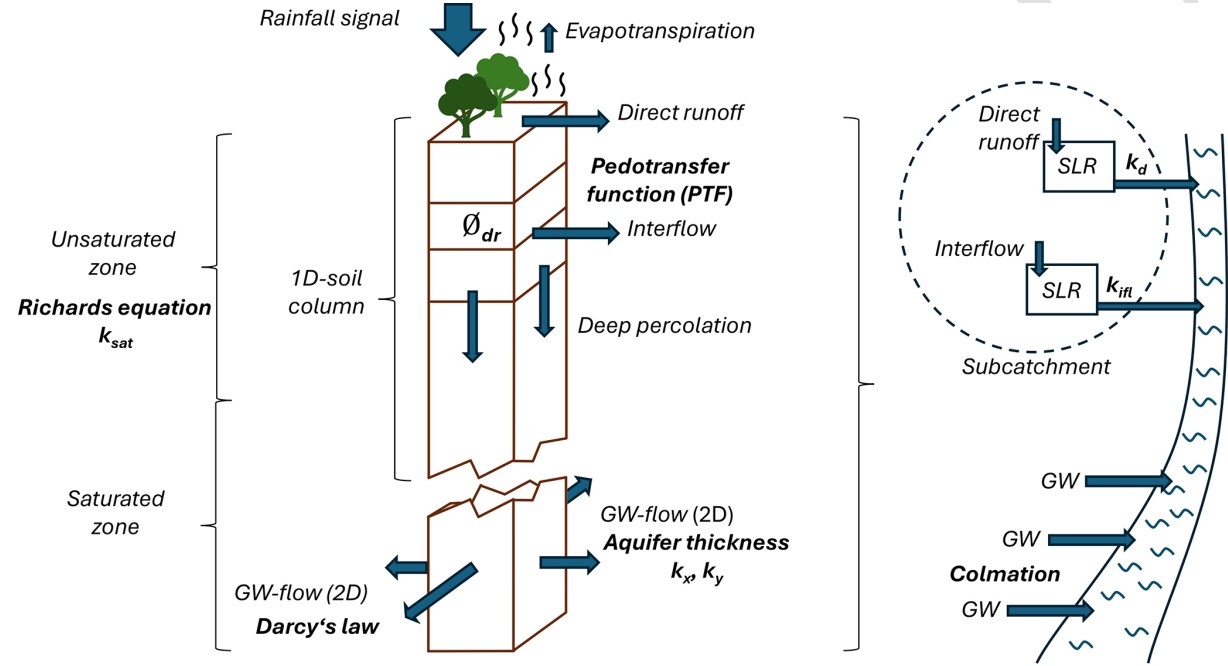

**Figure 2.** Conceptual diagram of the WaSiM model's structure. Bold text symbolises certain parameters or functions that are used to derive parameter values for the model parameterisation. Blue arrows indicate water fluxes within the model.

40 m spatial resolution showed to be the best trade-off between spatial resolution precision and model computation time. This also applies to the chosen temporal resolution of 1 h. INTERMET data (Gerlach, 2006) were used as input time series for meteorological data (temperature, precipitation, radiation, humidity). The data range from 1 January 2010–31 December 2020. Wind data were taken from the Agrarmeteorologie Rheinland-Pfalz (2024) for the stations Avelsbach [49.754° N, 6.693° E], Hermeskeil [49.655° N, 6.933° E], and Konz [49.687° N, 6.572° E]. Missing entries for periods of a few hours were manually resolved.

Following the preprocessing tool of WaSiM, TANALYS (Schulla, 2024b), the preprocessing tool of WaSiM, was used to calculate the required spatial information grids based on the digital elevation model. These spatial information grids include grids for the slope, exposition, subcatchments, river network, river width and depth, and colmation, as well as lateral aquifer conductivities ($k_x$ and $k_y$). A value of 50 was selected as the threshold for the river network. The threshold value describes from how many cells of runoff must be combined to form a water body cell in the model. Higher values for this threshold therefore result in a coarser river network,

while lower values result in finer river networks. The resulting network, based on the threshold value of 50 cells, showed the best fit with the water body of the catchment. Based on the soil types and land use information, profiles of the individual soils were created. These profiles contained data on thickness, soil type, depth, bulk density, carbonate content, humus content, and dry bulk density of the individual horizons.

Simulated soil hydraulic properties include hydraulic conductivity, soil water content at field capacity, and saturated water content. These are described using van Genuchten parameters and the saturated hydraulic conductivity $k_{sat}$. We used 12 different pedotransfer functions (PTFs) to calculate these parameter values. Pedotransfer functions can derive the required values for the van Genuchten parameters from measured soil data based on certain regression curves. Combinations of used pedotransfer functions are shown in Table 1. For the first seven PTF combinations, values for the saturated hydraulic conductivity $k_{sat}$ were taken from the KA5 Ad-hoc-AG Boden (2005). For PTF combinations 8–12, the values were calculated by the respective PTF's equation for $k_{sat}$. The chosen PTFs mainly differ in their underlying data, soil sample size, and considered soil parameters for the resulting predictive equations. Also, some PTFs are based on regular regression models while others are based on neural networks for deriving the hydraulic parameter values. A comprehensive analysis of the characteristics of PTFs 1–11 and their impact on the derived hydrological soil properties has been provided by Mohajerani et al. (2021). Each soil was then initialised with 27 layers, including a groundwater layer, and their respective hydraulic properties derived by the PTFs.

## 2.4 Calibration scheme

The calibration approach and its individual steps are described and summarised in Table 2. In Fig. 3, the individual calibration steps are depicted schematically in connection to the corresponding hydrological processes conceptualised in the WaSiM model structure. In step 1, evapotranspiration parameters are calibrated using MODIS evapotranspiration patterns. This step ensures a closed water balance as well as correct ETa patterns across different land uses. Step 2 adjusts the geometry and transmissivity of the groundwater model. In step 3, the rate of groundwater recharge via the amount of water entering the aquifer is calibrated. Both steps aim at correctly depicting the groundwater model behaviour with its contribution to total discharge. In step 4, the different PTFs are evaluated by comparing the patterns of dominant runoff processes under a synthetic heavy rainfall event. This step allows for the identification and exclusion of unsuitable PTFs that generate inaccurate runoff patterns. In step 5, the peaks in the hydrograph, represented as the high flow volume on the flow duration curve, are then adjusted to calibrate the model parts that are directly influenced by precipitation. Finally, in step 6, the model is evaluated in terms of its ability to predict

the overall discharge, based on hydrograph efficiency metrics in a split-sample test.

## 2.5 Calibration of ETa patterns (step 1)

The approach for calibrating the ETa patterns was originally described by Casper et al. (2023). According to this, the evapotranspiration parameters were calibrated using land use-specific MODIS-derived data (MOD16A2) and validated against Landsat-derived ETa data. This calibration step enhances the representation of spatiotemporal ETa dynamics within the model and closes the water balance at the catchment outlet. All ETa related parameters are taken from Casper et al. (2023).

## 2.6 Calibration of transmissivity (step 2)

Firstly, the model was calibrated in terms of its ability to reproduce the groundwater behaviour and the associated base flow. For this purpose, simulation runs were carried out with the initial parameterisations. A model run for the period from 1 January 2010–31 December 2014 served as a preliminary run for model spin-up, while the actual model run was then carried out for the period from 1 January 2010–31 December 2020 using the preliminary run as the initial model state.

We then examined the groundwater behaviour of the catchment and the model by applying the delayed flow index (DFI) method of Stoelzle et al. (2020) to the measured gauging data and the simulated hydrograph. For this, the series of discharge values of the hydrograph is divided into non-overlapping sections. These sections span a specific period of block-length $n$ (days) with $1 \le n \le 180$. The minimum flow value of each interval is then compared with the ones from adjacent intervals. If a minimum value multiplied by a specific factor $f = 0.9$ is smaller than the adjacent minima, a turning point (TP) is defined at its position. These TPs are then connected and form a delayed-flow hydrograph, which results in a specific hydrograph for each block length $n$. From this, the delayed-flow index (DFI) is calculated for each block length as the ratio of the sum of the delayed-flow to the sum of the total flow. An example how the applied block lengths result in different hydrographs can be seen in Fig. 4.

The DFI analysis was conducted using R (R Core Team, 2023) within RStudio (RStudio Team, 2020). The above-mentioned method was applied to the simulated hydrograph. DFI values for the individual block lengths $n$ were calculated using the function baseflow from the package *lfstat* (Gauster et al., 2022). The resulting DFI values for all block lengths $n$ were then plotted in a diagram, creating a characteristic delay curve (CDC). The *find_bps* function from the R-package *segmented* (Muggeo, 2008) was then used to determine the breakpoints of the curve. Breakpoints are defined as those points of the curve at which a change in the discharge characteristic can be determined (sudden change in slope).

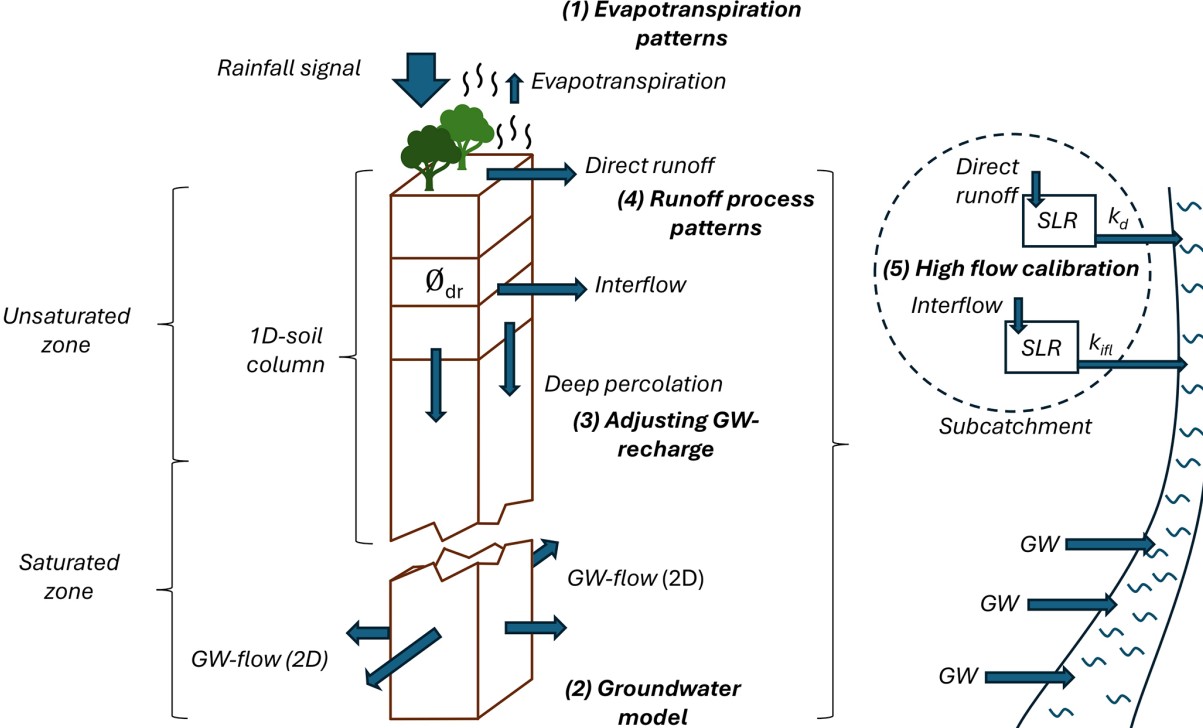

**Figure 3.** Conceptual diagram of the WaSiM model structure and the steps of the associated calibration approach. Evapotranspiration patterns are calibrated using MODIS evapotranspiration data (1). The groundwater model flow is then calibrated using the transmissivity (2). Groundwater recharge, i.e. the amount of water, is adjusted by calibrating the amount of interflow with the scaling factor $d_r$ (3). Dominant runoff process patterns derived from an extreme synthetic rainfall event are compared with the reference map to filter for matching patterns (4). Calibration of high discharge (peak flows) by adjusting the recession parameters of the direct runoff and interflow single linear reservoirs for each subcatchment (5). The last step, the evaluation of the hydrograph with efficiency metrics (6), is not shown in this concept figure.

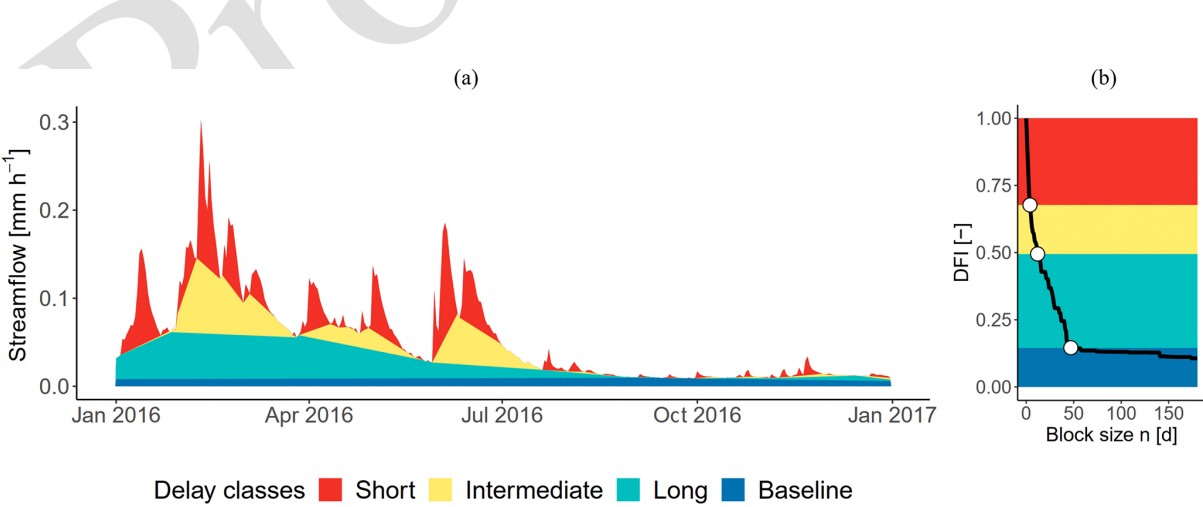

**Figure 4.** Application of the DFI approach. Panel **(a)** is the hydrograph separation according to calculated break point values for block lengths. The corresponding characteristic delay curve (CDC) derived from the hydrograph separation over all block lengths of $1 \leq n \leq 180$ is shown in **(b)**.

**Table 1.** PTF combinations used to estimate the van Genuchten parameters and the saturated hydraulic conductivities.

| PTF combination | Van Genuchten parameters | Soil hydraulic conductivity $k_{sat}$ |
| --- | --- | --- |
| 1 | Wösten et al. (1999) | Ad-hoc-AG Boden (2005) KA5 |
| 2 | Renger et al. (2008) | Ad-hoc-AG Boden (2005) KA5 |
| 3 | Weynants et al. (2009) | Ad-hoc-AG Boden (2005) KA5 |
| 4 | Zacharias and Wessolek (2007) | Ad-hoc-AG Boden (2005) KA5 |
| 5 | Teepe et al. (2003) | Ad-hoc-AG Boden (2005) KA5 |
| 6 | Zhang and Schaap (2017): Rosetta H2w | Ad-hoc-AG Boden (2005) KA5 |
| 7 | Zhang and Schaap (2017): Rosetta H3w | Ad-hoc-AG Boden (2005) KA5 |
| 8 | Wösten et al. (1999) | Wösten et al. (1999) |
| 9 | Renger et al. (2008) | Renger et al. (2008) |
| 10 | Zhang and Schaap (2017): Rosetta H2w | Zhang and Schaap (2017): Rosetta H2w |
| 11 | Zhang and Schaap (2017): Rosetta H3w | Zhang and Schaap (2017): Rosetta H3w |
| 12 | Szabó et al. (2021): euptfv2 | Szabó et al. (2021): euptfv2 |

**Table 2.** Scheme for the calibration and evaluation approach applied in this study.

| Step | Description | Aim | Scale | Behaviour |
| --- | --- | --- | --- | --- |
| 1 | Adjustment of ETa (for each land use type) | Close the water balance, match spatial patterns with MODIS | Spatial and temporal pattern match | Mean long-term behaviour |
| 2 | Adjusting GW model (transmissivity) | Calibrated baseflow within the DFI | Temporal match (DFI) | Mean long-term behaviour of GW submodel |
| 3 | Adjusting GW recharge | Partitioning GW/interflow | GW/interflow | Long-term GW recharge |
| 4 | Checking runoff generation processes | Match runoff processes with reference map (BHK) | Spatial match | Model behaviour test for extreme precipitation event (100 mm) |
| 5 | Adjusting high flows | Adjusting signature indices | Match on flow duration curve | Rainfall-fed part of the hydrograph |
| 6 | Final model evaluation | Peak flow statistics, split-sample test | Flow duration curve, hydrograph | Consistency at catchment outlet |

For this, $n_{LS} = 4$ linear segments were fitted to the CDC by residual minimisation, resulting in a total of $n_{BP} = 3$ breakpoints along the curve. The area between the last breakpoint ($n = 48$) and $n = 180$ was then considered as the area of the CDC where the aquifer's baseflow is the dominant contribution. This was the area where our groundwater model calibration took place. This procedure was then done for each PTF, resulting in a CDC for each PTF parameterisation.

Calibration was done to fit the slope of the rear area of the CDC. As the slope is determined by the transmissivity of the aquifer, adjustments were made for the model parameters $k_x$, $k_y$, and colmation, as well as the thickness of the aquifer. This was done until the slopes of the rear ends of the CDC for the simulations were identical with the slope of the CDC for the gauging station. A table with the calibrated model parameters can be found in the Appendix (Table B1).

### 2.7 Calibration of groundwater recharge (step 3)

After the groundwater transmissivity was adjusted, the different PTFs showed varying proportions in their CDC curves' rear areas. This indicated that the different PTFs lead to different amounts of water that reached the aquifer. To fit the simulation's CDC curve height to the height of the curve for the measured discharge, the value for the model parameter drainage density ($d_r$) was adjusted for each PTF independently. This conceptual parameter describes how much of the infiltrating water in the soil passes into the interflow and thus does not reach the aquifer. It therefore controls the amount of water contributing to groundwater recharge. As per Schulla (1997), the parameter $d_r$ is included in the formula for the interflow as:

$$q_{ifl} = k_{s(\Theta_m)} \cdot \delta z \cdot d_r \cdot \tan\beta, \tag{3}$$

with: $k_s$ being the saturated hydraulic conductivity [$m\,s^{-1}$]; $\Theta_m$ being the water content in the actual layer $m$ [–]; $d_r$ being

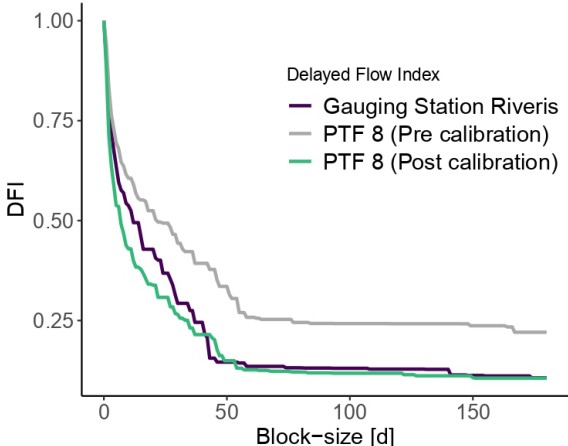

**Figure 5.** CDCs for the uncalibrated groundwater model and after groundwater model calibration, exemplarily for PTF 8.

the scaling parameter for the interflow to consider anisotropy of $k_{s,horizontal}$, compared to $k_{s,vertical}$; and $\beta$ being the slope angle with a maximum of $\beta = 45°$.

In this context, higher values of $d_r$ represent soil with stronger lateral drainage capabilities. This usually leads to more interflow and therefore less water that can infiltrate into the aquifer and contribute to groundwater recharge. Regarding the groundwater recharge calibration, higher values for $d_r$ lowered the curve, especially in the rear end. This brought the DFI values into the range of the reference curve (Fig. 5) for PTFs that initially showed higher CDCs in the rear area. For CDCs of PTFs that were lower than the reference CDC of the gauging station, the value for $d_r$ had to be lowered. This reduced interflow and increased the groundwater recharge. A table with the values of $d_r$ for the different PTFs can be found in the Appendix (Table B2).

## 2.8 Evaluation of dominant runoff process patterns (step 4)

In the next step, the different PTFs were compared regarding their ability to accurately depict the surface runoff processes in the catchment area under a heavy precipitation event. This step served to filter out those PTFs that are not capable of simulating the correct runoff patterns. For this purpose, the approach developed by Mohajerani et al. (2023) for comparing the runoff processes was used and adapted for our calibration scheme.

The soil hydrological map (BHK) of Rhineland-Palatinate from Steinrücken and Behrens (2010) was used as a reference for our comparison. The BHK is a map that depicts which runoff type dominantly appears under a heavy precipitation event. It divides the runoff into saturated overland flow (SOF), subsurface flow (SSF), and deep percolation (DP). Two finer classifications for SOF and SSF are characterised by different delay times. However, the WaSiM model does

not consider the delay but only the runoff type itself. Therefore, we only used the three main groups and not the subgroups for the comparison. We also refrained from subdividing the model processes according to the fractions, as suggested by Mohajerani et al. (2023). This was done because the soil hydrological map categorises the subclasses according to the delay and not to the proportions of runoff processes. A division by fractions therefore would not be fully comparable with a division by delay times (as in the BHK).

The BHK was adjusted to the Riverisbach catchment boundaries and rasterised to a resolution of $40\,m \times 40\,m$. This was done to facilitate a direct comparison between simulated runoff processes and the BHK as reference. For the comparison, the model state at the end of 31 December 2014 was used as the initial state of this step's model run. This initial state was then used to carry out a 7 d run-up under controlled climatic conditions (temperature $= 10°C$, radiation $= 0\,W\,m^{-2}$, wind speed $= 0\,m\,s^{-1}$, relative humidity $= 100\%$, and precipitation $= 0\,mm$) for the entire duration. This was done to eliminate influence of melting snow on the runoff analysis during the following main run as well as bringing soil moisture to field capacity. The final state of this preliminary run then served as the initial state for another 7 d model run. During this run, the catchment was irrigated with 100 mm of rain over the first seven hours ($14.286\,mm\,h^{-1}$). Over the simulation period of these seven days, the cumulative runoff fractions for each cell of the catchment grid were calculated. From the calculated fractions of runoff per grid cell, maps were created where each grid cell's dominant runoff process was attributed to. This resulted in a dominant runoff process map for each PTF.

The simulated runoff process patterns were then compared with the runoff process patterns of the BHK. For this purpose, the comparison approach using the spatial efficiency metric (SPAEF) (Stisen et al., 2017; Demirel et al., 2018) was adapted. The SPAEF is to be understood as a measure of spatial similarity. It is defined as:

$$SPAEF = 1 - \sqrt{(\alpha - 1)^2 + (\beta - 1)^2 + (\gamma - 1)^2} \qquad (4)$$

$$\alpha = \rho(A, B) \qquad (5)$$

$$\beta = \left( \frac{\sigma_A}{\mu_A} \middle/ \frac{\sigma_B}{\mu_B} \right) \qquad (6)$$

$$\gamma = \frac{\sum_{j=1}^{n} \min(K_j, L_j)}{\sum_{j=1}^{n} K_j}, \qquad (7)$$

with $\alpha$ being the Pearson correlation coefficient between the simulated grid ($A$) and the reference grid ($B$). $\beta$ is the fraction of coefficient of variations as an indicator of spatial variability. $\gamma$ is the percentage of histogram intersection (Demirel et al., 2018). The closer the SPAEF value is to 1, the higher the similarity between the compared patterns. During our analysis, however, we encountered a limitation with the standard SPAEF formula when applied to patterns consisting of only three groups. Specifically, the Pearson correlation

coefficient, as a component of the SPAEF, tended to yield lower values if deviations occurred in marginal areas. This occurred even when there was substantial overall agreement. To address this issue, we adapted the SPAEF calculation by substituting the Pearson correlation component. Instead, we used a direct measurement of percentage agreement between the simulation and the reference map grids. This adjustment led to the development of a modified SPAEF formula:

$$SPAEF_{mod} = 1 - \sqrt{(\delta - 1)^2 + (\beta - 1)^2 + (\gamma - 1)^2} \qquad (8)$$

$$\delta = \frac{\sum_{j=1}^{n_g} 1}{n_g} \quad \text{for} \quad A_j = B_j, \qquad (9)$$

where $\delta$ is the percentage match of all grid fields between simulated map ($A$) and reference map ($B$). It is calculated as the fraction of the amount of identical grid cell pairs between both maps to the number of grid cells in one map ($n_g$). $\beta$ and $\gamma$ remain unchanged. This new equation for $SPAEF_{mod}$ allowed us to correctly analyse the agreement between the simulated runoff patterns and the reference patterns of the hydrological map (BHK). A separate $SPAEF_{mod}$ value was then calculated based on the dominant runoff process map for each PTF.

### 2.9 Calibration of high flow discharge (step 5)

The discharge peaks of the model were calibrated by adjusting the coefficients of the single linear reservoirs for the direct runoff ($k_d$) and the interflow ($k_{ifl}$). The metrics of the signature indices (Casper et al., 2012) were used to evaluate the calibration of the individual linear reservoirs. These indices consider different sections and properties of the flow duration curves (FDCs) of simulated and measured discharge and compare them against each other. This yields a percentage bias for each signature index parameter. The *BiasRR* describes the percent bias in the mean values. The *BiasFD-Cmidslope* describes the percent bias in slope of the mid-segment. The *BiasFHV* describes the percent bias in high-segment volumes (upper 2 %). The *BiasFLV* is the difference in the long-term baseflow. The *BiasFMM* depicts the percent bias in mid-range flow levels.

First, the coefficient for the direct runoff single linear reservoir, $k_d$, was calibrated. A low value of 2 seemed to fit best for most PTFs, as the proportion of direct runoff in the total runoff was low and did not need to be delayed any further. For some PTFs, where the fractions of direct discharge were higher, the value for $k_d$ had to be increased. The value of *BiasFHV* was then minimised by adjusting the coefficient for the interflow runoff single linear reservoir, $k_{ifl}$. This was done to adjust the peaks of the simulated hydrograph to more closely resemble those of the measured hydrograph of the catchment. Higher values for $k_{ifl}$ lead to a stronger delay of the interflow runoff. This results in lower peaks of the discharge.

### 2.10 Final model evaluation (step 6)

#### 2.10.1 Characteristic delay curve (CDC) comparison

The CDCs for the different PTFs were compared to determine how well the discharge is simulated in the interflow area. For this purpose, the Manhattan distance (MHd) between the CDCs between $n = 1$ and $n = 43$ (last breakpoint of the measured data) was calculated according to the following formula:

$$d(A, B) = \sum_{i=1}^{n} |A_i - B_i|, \qquad (10)$$

where $A$ represents the values of the CDC for the gauging station and $B$ the values for the curve of the simulation.

#### 2.10.2 High discharge histogram overlap (HDHO) analysis

In addition, a high discharge histogram overlap (HDHO) analysis was carried out based on the hydrographs. By comparing the histograms of the temporal peak discharge distribution for the simulated and measured hydrograph, the model's capability of simulating the strongest discharge events can be assessed. For this purpose, the maximum discharge value of each year was determined. This was done for each PTF's hydrograph and for the measured data. The data were plotted in a histogram. The histogram overlap between simulated and measured data were then calculated for each PTF according to following formula:

$$HDHO = \frac{\sum_{j=1}^{n} \min(K_j, L_j)}{\sum_{j=1}^{n} K_j}, \qquad (11)$$

where $n$ is the number of bins, $K_j$ the number of values within bin $j$ for the reference (gauging station), and $L_j$ the number of values in bin $j$ for the simulation. This was done to determine a measure of the predictive accuracy of the discharge peaks. High histogram overlap values indicate a model's better predictive accuracy. Lower values represent poorer model capabilities of high discharge prediction.

#### 2.10.3 Hydrograph efficiency metrics

The hydrographs of the final simulations were then compared with the measured hydrograph by applying a split sample test. This was done to evaluate the model's ability to correctly predict the overall discharge. For this purpose, three metrics were chosen. These include the Kling–Gupta efficiency (KGE) to evaluate the correspondence between observed and simulated hydrographs. It considers aspects like correlation, bias, and variability (Kling et al., 2012). The Nash–Sutcliffe model efficiency coefficient (NSE) was used to evaluate how well simulated and measured values fit the 1 : 1 line (Nash and Sutcliffe, 1970). It puts a special focus on the prediction

of correct volume. The third metric included was the PBIAS (percent bias). This metric is a measurement of the average tendency of the simulated data to be larger or smaller than their observed counterparts (Gupta et al., 1999). All three efficiency metric values were calculated for the calibrated model hydrographs for each PTF.

## 2.11 Evaluation of PTF suitability

For the evaluation of the different PTFs, the respective model performance for each calibration step was evaluated. In order to be considered as satisfactory regarding the groundwater model calibration, the PTF must allow the model's CDC to be adjusted in slope and height to match the reference curve of the gauging station. If the slope or the height could not be brought into concordance with the reference, the PTF was considered as unsatisfactory. For the evaluation of the dominant runoff process patterns, the respective SPAEF$_{\text{mod}}$ values were used as the discriminatory statistic. Here, all PTFs that lead to SPAEF$_{\text{mod}}$ values above 0.5 were considered as satisfactory. The threshold was chosen as values above 0.5 usually lead to already well-fitting patterns (Mohajerani et al., 2023). For the evaluation of the discharge prediction, the NSE, KGE and PBIAS of the validation period are used as the discriminatory variable. PTFs are considered satisfactory when the PBIAS is within a range of $\pm 10.0\%$ and the NSE and KGE are above 0.7. Other studies already consider values of 0.5 as satisfactory for NSE or KGE (Moriasi et al., 2015; Rogelis et al., 2016). However, we aimed for a model that shows stronger congruence with the reference discharge curve, therefore choosing a higher threshold value. The threshold for the PBIAS was also set stricter, as others already define values between $\pm 25\%$ as very good (Moriasi et al., 2007). Then, an overall benchmark was deducted based on the three individual evaluation results. A PTF was then only considered satisfactory if it lead to satisfactory results for all three evaluation steps.

## 3 Results

### 3.1 ETa patterns (step 1)

In step 1, we were able to use the already parameterised and calibrated values for the ETa-relevant plant properties from Casper et al. (2023). This made a separate evaluation of calibrated parameter values obsolete. The adequacy of the used values was also supported by the closed water balance in our model (see Sect. 3.4), with deviations ranging from $-8.37\%$ to $-0.04\%$.

### 3.2 Groundwater model parameterisation (step 2 and 3)

The evaluation of the groundwater model adjustment (Fig. 6) shows that, in step 2 of our approach, we successfully matched the slope of the CDC to the observed data for all PTFs. This was achieved by using a single layer aquifer with a thickness of 1 m and lateral hydraulic conductivities of $3 \times 10^{-5}\,\text{m}\,\text{s}^{-1}$. In step 3, the CDC height could also be adapted to the course of the gauging station curve for almost all PTFs except PTF 9 and 10. The corresponding calibrated values for $d_{\text{r}}$ range from 6 for PTF 4 up to 60 for PTF 2, In the front part of the curve, the simulations almost exclusively run below the reference curve of the gauging station.

### 3.3 Dominant runoff process patterns (step 4)

In step 4, the simulated dominant runoff processes for each PTF were compared to the reference map (BHK) to evaluate how well each PTF represents the spatial patterns of runoff (see Fig. 7). The overview of the simulated runoff processes shows that some PTFs deviate significantly from the reference map. Except for PTFs 4, 9, and 10, all show dominant interflow over most of the catchment area. PTFs 1, 2, 3, and 12 show hardly any significant areas of deep percolation. However, in the reference map of the BHK, deep percolation can be found in the northern and southern edges of the catchment. Only PTFs 5, 6, 7, and 11 show such areas with dominating deep percolation at the same positions as the BHK. PTF 4 shows almost exclusively dominant, extensive surface runoff. It only shows interflow around the watercourse. This differs highly from the reference map. In comparison, PTF 9 and 10 show strongly dominating deep percolation over a large area. Also, only narrow areas with interflow can be found in the vicinity of the watercourse. The area with surface runoff in the west is also not depicted correctly in both PTFs. For all PTFs, the high correspondence between simulated and reference map for the direct runoff patterns results from the fact that, by definition, surface runoff occurs in the model when a watercourse flows through a cell.

The overall values as well as the individual metrics of the SPAEF$_{\text{mod}}$ metric are listed in Table 3. The SPAEF$_{\text{mod}}$ values summarise the values for the three individual parameters. PTFs 3 and 5 achieve high values of just over 0.75. Their simulated patterns for these PTFs therefore show high similarity to the patterns of the reference map. PTFs 1, 2, 7, 8, and 12 show values in the mid-range. They show strong overall similarities between the patterns, while individual areas are not correctly depicted in the simulated patterns. PTFs 4, 6, 9, 10, and 11 have the lowest values. They are all below 0.

### 3.4 High flow calibration (step 5)

The signature indices, including an evaluation of the high discharge (step 5), show a pronounced amplitude across the range of PTFs for some indices. For the *BiasRR*, which represents the mean deviation and thus the water balance, most PTFs show only small deviations of around 5 %. Only PTFs 4 and 5 have higher deviations of close to 10 %. It is striking that most PTFs underestimate the water balance, i.e. show

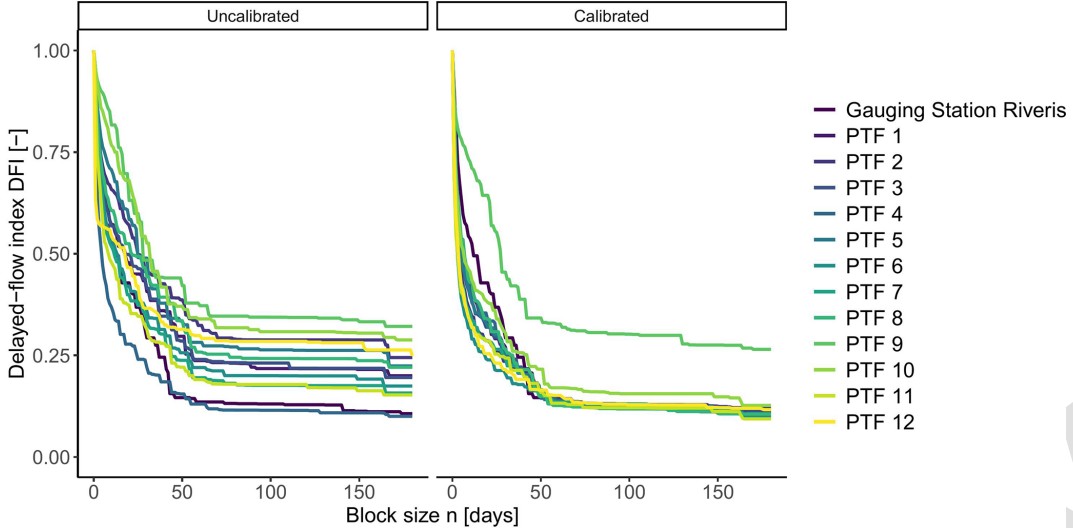

**Figure 6.** CDCs for the uncalibrated groundwater model and after groundwater model calibration for each PTF.

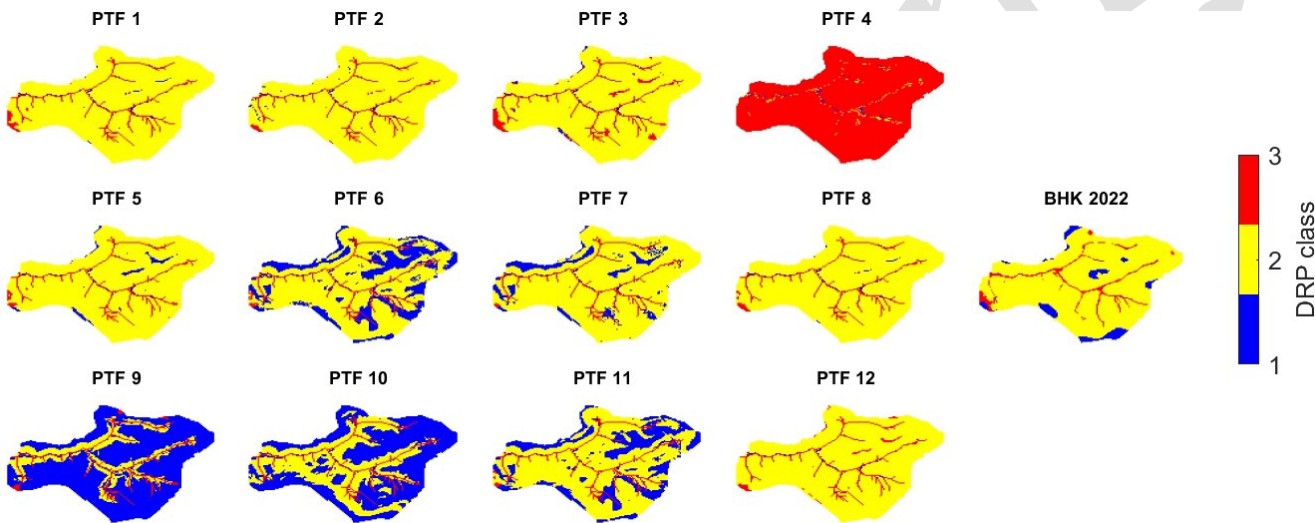

**Figure 7.** Spatial patterns for the simulated dominant runoff processes and the corresponding BHK reference map after a synthetic rainfall event.

negative deviations. Only PTF 7 has a value of almost 0 % and therefore shows no over- or underestimation. The *biasFDCmidslope*, which describes the reactivity of the hydrograph, shows a large amplitude. PTFs such as 1, 2, 3, 4, 5, 10, and 11 show deviations of well below 10 %. PTF 6 shows an upward deviation of 21.01 %. PTF 9 shows a downward deviation of −30.49 %. Almost all PTFs show a *BiasFHV* close to 0. Only PTF 9 shows significant deviation of −31.49 %. Most PTFs show a moderate underestimation of between −10 % and −15 % for the *BiasFLV*. Only PTF 9 shows a considerable upward deviation of 41.49 %. The deviation of the median (*BiasFMM*) shows a strong amplitude across the various PTFs. PTF 6 shows the largest negative deviation of −29.62 %. PTF 9 shows the largest positive de-

viation of 28.79 %. PTF 3 has the lowest deviation from zero at just −1.04 %.

## 3.5 Final model evaluation (step 6)

The Manhattan distances, calculated between the CDCs of simulated and observed data across the range of $n$ values from $n = 1$ to $n = 43$, show considerable variabilities across all PTFs (Table 5). While PTF 10 has a distance value of only 2.01, the distance value of PTF 9 is several times higher with 6.68. PTFs 1 and 8 also show small distances, while the other PTFs are located in the middle range. For the high discharge histogram overlap (HDHO), PTF 4 shows the lowest value of

**Table 3.** Metrics for the comparison of simulated dominant runoff processes and the BHK reference map.

| PTF | % match | $\alpha$ | Histogram overlap | $SPAEF_{mod}$ |
|---|---|---|---|---|
| 1 | 0.89 | 0.74 | 0.95 | 0.72 |
| 2 | 0.88 | 0.74 | 0.95 | 0.71 |
| 3 | 0.88 | 0.83 | 0.96 | 0.76 |
| 4 | 0.06 | 0.28 | 0.08 | −0.49 |
| 5 | 0.89 | 0.83 | 0.97 | 0.79 |
| 6 | 0.71 | 1.94 | 0.79 | −0.01 |
| 7 | 0.82 | 1.20 | 0.96 | 0.73 |
| 8 | 0.89 | 0.73 | 0.95 | 0.71 |
| 9 | 0.27 | 5.38 | 0.31 | −3.49 |
| 10 | 0.45 | 3.51 | 0.50 | −1.62 |
| 11 | 0.71 | 1.96 | 0.79 | −0.03 |
| 12 | 0.88 | 0.73 | 0.95 | 0.71 |

**Table 4.** Signature indices of the calibrated model for different PTFs.

| PTF | BiasRR | BiasFDCmidslope | BiasFHV | BiasFLV | BiasFMM |
|---|---|---|---|---|---|
| 1 | −5.31 | 4.05 | 1.25 | −11.84 | −7.75 |
| 2 | −3.79 | 1.29 | −0.88 | −9.76 | −4.61 |
| 3 | −6.56 | −0.03 | −0.89 | −9.15 | −1.04 |
| 4 | −9.30 | 5.83 | −0.76 | −9.70 | −8.23 |
| 5 | −8.37 | 8.68 | −3.5 | −11.07 | −11.89 |
| 6 | −4.46 | 21.01 | 1.95 | −13.23 | −29.62 |
| 7 | −0.04 | 20.99 | 0.14 | −8.79 | −7.36 |
| 8 | −5.52 | 10.64 | 0.08 | −14.56 | −17.54 |
| 9 | −5.38 | −30.49 | −31.49 | 41.49 | 28.79 |
| 10 | −4.94 | 0.40 | −4.39 | −4.18 | −2.35 |
| 11 | −3.65 | 7.26 | 3.93 | −9.41 | −13.36 |
| 12 | −5.00 | 9.32 | 0.22 | −13.36 | −14.99 |

0.5. PTFs 1 and 12 show a high value of 0.9. Values of the other PTFs are located within 0.5 and 0.9.

The split-sample test carried out based on the simulated and measured hydrograph (Fig. 8) shows strong consistency with evaluation metrics of the model for the best parameterisation (PTF 8). The model shows high values for the efficiency measures for both the calibration and the validation period. Between calibration and validation, there is only a slight decrease in the NSE from 0.87–0.78, while the KGE decreases only minimally from 0.89–0.87. Values for the PBIAS slightly improve from around −5.52 % for the calibration period to 3.16 % for the validation period. Efficiency measures for the split-sample test of other PTFs (Table 5) show a large value range. For example, PTF 1 also shows relatively high values for the NSE and KGE. However, PTFs 4, 6, 9, 10, and 11 show low values. All other PTFs show values in between. The PBIAS shows values of around −5.00 % for most PTFs for the calibration period, while the values for the validation period are between −5 % and 5 % for all PTFs except PTF 7.

The hydrograph simulated by PTF 8 successfully replicates the measured hydrograph, with only slight underestimation of peak flows and a minor delay in response around December 2017. The model tends to smooth out finer fluctuations, resulting in a lower reactivity compared to observed data. Overall, however, PTF 8 closely mirrors the complex shape of the observed hydrograph. Hydrographs for other PTFs can be found in the appendix as Figs. A1 and A2.

The long-time discharge can also be depicted as a flow duration curve (Fig. 9). The flow duration curve for PTF 8 shows very good agreement in the high discharge volume. This corresponds to the discharge peaks of the hydrograph. In the middle part, the flow duration curve shows a kink. From there, it is no longer fully congruent with the curve for the measured discharge in areas for lower discharge volumes. The simulation slightly deviates from the measured flow duration curve in the area of very low discharges. However, it should be noted that the representation is logarithmic. The deviations occurring in the low discharge range therefore only account for a small proportion of the total discharge. PTF 8 therefore fits the flow duration curve of the reference the best. The other PTFs are deviating around the measured curve. Some overestimate the corresponding proportions and others underestimate the proportions. In the middle range, the results of the simulations are almost exclusively lower than the reference.

### 3.6 Overall evaluation of PTFs

The evaluation of all PTFs for the individual calibration steps shows that only two (PTFs 1 and 8) of the 12 PTFs used yield satisfactory results for all three calibration steps (Table 6). The majority of PTFs show satisfactory results for the calibration of the groundwater model. For the runoff process patterns, an increasing number of PTFs already show that they do not lead to satisfactory results. For the discharge prediction, only two of the PTFs used show acceptable results.

## 4 Discussion

This study employed a multi-step calibration approach designed to incrementally improve the accuracy of hydrological simulations by systematically targeting specific components of the water balance model. The following paragraphs discuss the results of each calibration step in detail.

### 4.1 Evapotranspiration/water balance (step 1)

We used calibrated vegetation parameters from Casper et al. (2023). Because of the almost closed water balance (*BiasRR* in Table 4), an additional calibration step for evapotranspiration parameters was not necessary in our case. Only if the water balance could not be closed at the catchment outlet would it have been necessary to adjust the evapotranspiration parameters.

**Table 5.** Efficiency metrics for the calibrated model for different PTFs.

| PTF | MHd | HDHO | $NSE_{cal}$ | $KGE_{cal}$ | $PBIAS_{cal}$ | $NSE_{val}$ | $KGE_{val}$ | $PBIAS_{val}$ |
|---|---|---|---|---|---|---|---|---|
| 1 | 3.19 | 0.9 | 0.86 | 0.89 | −5.31 | 0.78 | 0.87 | 2.64 |
| 2 | 4.88 | 0.7 | 0.66 | 0.82 | −3.79 | 0.59 | 0.79 | 1.28 |
| 3 | 3.59 | 0.8 | 0.72 | 0.84 | −6.56 | 0.63 | 0.81 | −0.15 |
| 4 | 3.36 | 0.5 | 0.67 | 0.79 | −9.30 | 0.45 | 0.69 | −5.25 |
| 5 | 3.68 | 0.7 | 0.72 | 0.83 | −8.37 | 0.55 | 0.75 | −3.16 |
| 6 | 6.60 | 0.6 | 0.49 | 0.73 | −4.46 | 0.34 | 0.70 | 5.01 |
| 7 | 5.04 | 0.8 | 0.60 | 0.81 | −0.04 | 0.49 | 0.77 | 8.67 |
| 8 | 2.76 | 0.8 | 0.87 | 0.89 | −5.52 | 0.78 | 0.87 | 3.16 |
| 9 | 6.68 | 0.8 | 0.54 | 0.61 | −5.38 | 0.49 | 0.60 | −3.24 |
| 10 | 2.01 | 0.6 | 0.59 | 0.78 | −4.94 | 0.47 | 0.75 | 3.16 |
| 11 | 5.36 | 0.8 | 0.55 | 0.76 | −3.65 | 0.43 | 0.74 | 5.39 |
| 12 | 5.84 | 0.9 | 0.70 | 0.83 | −5.00 | 0.64 | 0.80 | 0.64 |

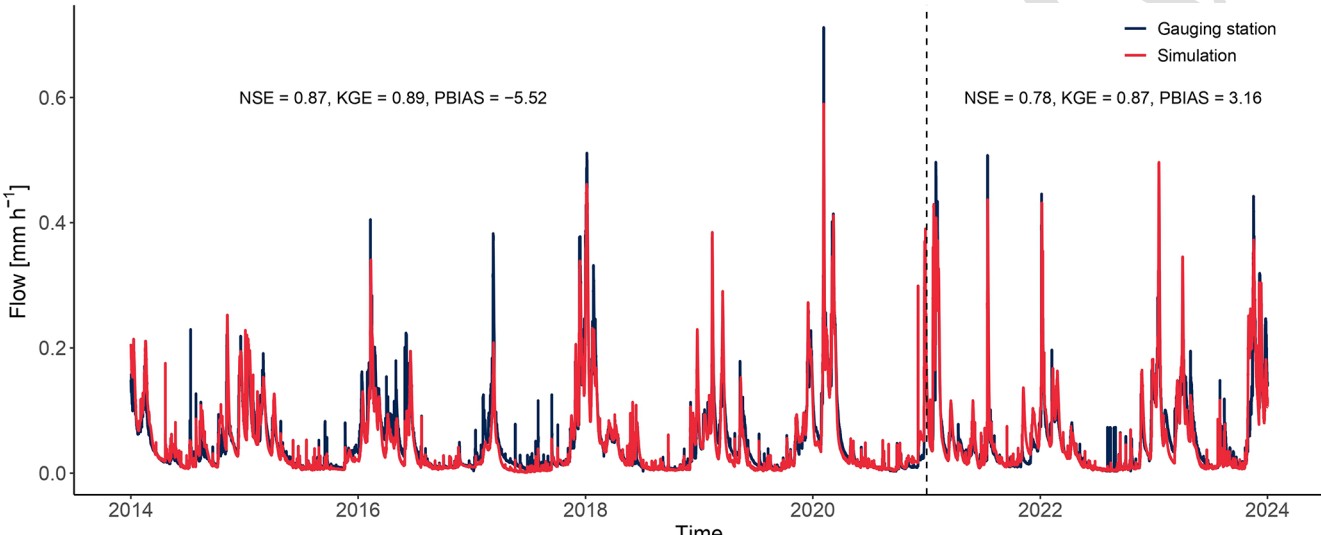

**Figure 8.** Measured (gauging station) and simulated (PTF 8) hydrographs. Period before the dashed vertical line is the calibration period, while the one right of the dashed line marks the validation period. Efficiency metric values are shown for their respective period.

## 4.2 Groundwater model (steps 2 and 3)

Fitting to the characteristic delay curve (CDC) is a suited method for the calibration of the groundwater model in terms of its mean long-term behaviour (Fig. 6). The gradient of those segments of the CDCs which correspond to longer delay intervals (higher $n$ values) are highly sensitive to aquifer transmissivity parameters ($k_x$, $k_y$ and thickness). On the other hand, the long-term groundwater recharge depends on the interflow intensity, which is adjusted by the parameter drainage density $d_r$. This approach effectively modified the height of the CDCs across most PTFs. However, two PTFs (PTFs 9 and 10) did not allow a good adjustment to the observed CDC height, due to lack of soil stratification in their parameterisation. These two PTFs estimate the hydraulic properties based on grain size, while key factors like depth or bulk density – typically considered in other PTFs

or when using the KA5 standard for saturated hydraulic conductivity ($k_{sat}$) – are not addressed. This means that, in the absence of stratification, there is little interflow and a large portion of water percolates into the aquifer (Ahuja et al., 1981). Without stratification, interflow cannot be controlled by the scaling factor $d_r$ because there is too little interflow to begin with. The consistent underestimation of the initial segments of the CDCs suggests that the catchment is delaying certain parts of the water more than the model does (Yeh and Chen, 2022). This could theoretically be resolved by increasing the interflow delay through increasing values for $k_{ifl}$. However, as our catchment is mainly interflow dominated, the discharge peaks are almost exclusively interflow. Such an adjustment could reduce peak discharge significantly, which might compromise the hydrograph fit, as noted by Shrestha et al. (2013). Therefore, we assume that a two-layer aquifer model with distinct transmissivities would probably better

**Table 6.** Evaluation of the model based on different PTF parameterisations for the three main calibration steps. A mark indicates satisfactory results for the respective step. A mark for overall benchmark is granted if all three calibration steps are marked as satisfactory.

| PTF | Groundwater model calibration | Runoff process patterns | Discharge prediction | Overall benchmark |
|---|---|---|---|---|
| 1 | × | × | × | × |
| 2 | × | × | – | – |
| 3 | × | × | – | – |
| 4 | × | – | – | – |
| 5 | × | × | – | – |
| 6 | × | – | – | – |
| 7 | × | × | – | – |
| 8 | × | × | × | × |
| 9 | – | – | – | – |
| 10 | – | – | – | – |
| 11 | × | – | – | – |
| 12 | × | × | – | – |

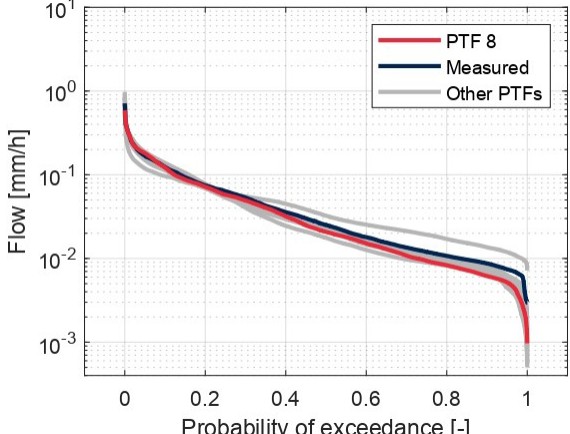

**Figure 9.** Flow duration curve for the gauging station for the simulation with PTF 8 (red) and the other PTFs (grey).

represent the complex groundwater dynamics in our catchment.

### 4.3 Evaluation of dominant runoff processes (step 4)

The evaluation of dominant runoff processes has shown that most PTFs can reproduce the pattern of the reference with reasonable accuracy (Fig. 7). However, PTFs 4, 6, 9, 10, and 11 showed significant deviations from the reference patterns, which indicate that these PTFs produce soil parameter estimates that differ substantially from actual field conditions. This results in either little interflow and too much surface runoff (PTF 4) or too much deep percolation and little interflow (PTFs 9 and 10). The high proportion of surface runoff and low fractions of interflow of PTF 4 are probably due to the low hydraulic conductivities compared to other PTFs (Mohajerani et al., 2021). Therefore, the upper soil layers in the model quickly saturate during the synthetic rainfall

event which results in a predominance of surface runoff. In contrast, PTFs 9 and 10 lead almost exclusively to dominant deep percolation. This is due to a lack of soil stratification: only the grain size distribution is considered, while other properties such as bulk density or depth are neglected for the estimation of soil hydraulic conductivities (Renger et al., 2008; Zhang and Schaap, 2017). Consequently, the model assumes uniform permeability that allows most precipitation to infiltrate directly into the groundwater reservoir and bypass interflow pathways. However, the strong deviations in runoff pattern among these three PTFs can be systematically identified using the SPAEF$_\text{mod}$ metric. While the majority of PTFs achieved SPAEF$_\text{mod}$ values exceeding 0.65, which indicates good alignment with the reference map, PTFs 4, 6, 9, 10, and 11 showed significantly lower (in all cases, negative) values. This evaluation step serves as a reliable means to screen out PTFs that fail to capture dominant runoff processes accurately. This ensures that only soil parameterisations consistent with observed runoff fractions are considered in the final model selection process.

### 4.4 High flow calibration (step 5)

The subsequent adjustment of the rainfall-fed part of the hydrograph, e.g. discharge fractions in the high volume based on the signature indices (Table 4), showed good applicability. For all PTFs except 9 and 10, the *BiasFHV* could be brought close to zero. The water distribution could be shifted from peak discharge values towards mid-range discharge levels by adjusting $k_\text{d}$ or $k_\text{ifl}$. PTFs 9 and 10 lack volume in the discharge peaks due to the large proportion of water that infiltrates very quickly into the aquifer. Therefore, hardly any direct runoff or interflow is present, which could contribute to high volume discharge (Seiler and Gat, 2007). This is also reflected in the patterns for the dominant runoff processes. In that case, the parameter $k_\text{ifl}$ could not be used to shift more

water from the peaks to the stronger delayed portions of discharge without losing a significant amount of water volume in the peaks. This is probably because our study area produces only little direct runoff, the contribution of which to the total runoff is delayed via $k_d$. Therefore, mainly interflow contributes to the discharge. As a result, the hydrograph peaks in our model primarily reflect fast interflow rather than a balanced combination of direct runoff and interflow runoff. An independent adjustment, via $k_d$ and $k_{ifl}$, would only be possible if both runoff types are present to a certain extend. Adding a second aquifer layer with slightly higher conductivities than our current aquifer would enable us to represent a less delayed groundwater discharge that currently is depicted through interflow. As a result, less interflow would be needed to represent parts of the slow components and therefore could be used to model part of the peak discharge. However, the necessity of this depends entirely on the catchment characteristics (Natkhin et al., 2012; Kraller et al., 2014) and can be derived from a repeated application of the characteristic delay curve (Step 2 and 3).

## 4.5 Final model evaluation (step 6) and PTF evaluation

The hydrograph of the best fitting model (based on PTF 8) shows that the model is capable of correctly predicting the discharge (Fig. 8). This is also supported by high values of efficiency measures such as NSE (0.78), KGE (0.87), and a low PBIAS (3.16 %) for the validation period in the split-sample test. In addition, a high discharge histogram overlap (0.8) shows a good agreement in the peak discharge over time. However, the various PTFs show considerable deviations from each other. The choice of the pedotransfer function has a significant influence on the individual processes depicted by the model and therefore the correct choice of the pedotransfer function is crucial to develop a process-behaviourally correct model parameterisation. This is also consistent with the findings of Mohajerani et al. (2021) and Paschalis et al. (2022). Our multi-criteria calibration framework, with its combination of parameterisation steps, proved effective both in evaluating PTFs and refining the calibration itself. Inconsistencies with both the CDCs and the patterns of dominant runoff processes proved the non-suitability of PTFs 9 and 10. Likewise, PTF 4 was found unsuitable due to deviations in runoff process patterns, despite its potential for further groundwater volume adjustments via drainage density $d_r$. This shows that a holistic view of the different processes is indeed necessary, as one PTF can be suited for a single process such as the groundwater flow but unsuited for other processes.

A great advantage of our developed approach is the relatively simple applicability of the developed methods as well as the shown high selectivity regarding different calibrations and the selection of the most suitable one. It has been shown several times that the parameterisation of the soil properties is crucial for the hydrological behaviour of an area (Kubát

et al., 2024). However, the choice of the best suiting PTF is still given too little attention in hydrological modelling (Hmaied et al., 2024). Our approach allows the hydrological model to be parameterised with the most suitable PTF by both adjusting the aquifer parameterisation and evaluating the dominant runoff process patterns to filter out non-fitting PTFs. This is something that has not been incorporated into calibration approaches until now. In addition, information on aquifer properties is often lacking, which is why their correct parameterisation and calibration are often neglected in the calibration strategies for hydrological models (Ntona et al., 2022). However, our approach makes it possible to obtain information on aquifer behaviour from information that is usually available like the hydrograph. The gathered data can then be used for model calibration. This enables the correct representation of this discharge contributing process, i.e. the base flow generation, in the model.

## 4.6 Transferability and outlook

Our calibration approach is effectively transferable to other hydrological models and catchments, provided the necessary input parameters are available. For the first step, the calibration of ETa, remotely sensed ETa data are necessary. Here, readily available MODIS data can be used. Additionally, the application of the delayed flow index (DFI) requires only simulated and measured hydrographs, alongside a mechanism for adjusting groundwater recharge by percolating water. Models must support runoff partitioning into surface runoff, interflow, and deep percolation (groundwater recharge) to utilise the dominant runoff process comparison. For this, a reference for the spatial patterns, for example, the soil hydrological map, is necessary. While certain methods necessitate only discharge data, we emphasise the benefits of incorporating multiple evaluation approaches. This comprehensive parameterisation captures the catchment behaviour across various hydrological processes more accurately. Consequently, our methodology demonstrates broad applicability for future parameterisations of hydrological water balance models, particularly those with a process representation similar to the WaSiM model.

For SWAT+, for example, our approach could be adapted and used for a more process-behavioural focused calibration than the widely used calibration based on gauging data alone. For the calibration of the aquifer, we recommend using *gwflow* (Bailey et al., 2020) together with SWAT+, which allows for a more complex representation of aquifer behaviour in the model than SWAT+ alone does. Our approach using the DFI can then be applied exactly as described. It is also possible to evaluate the model with regard to the runoff components by comparing it with a reference map. For example, a tool such as *FieldSWAT* (Pai et al., 2012) could be used to record the spatial distribution of surface runoff, interflow or deep percolation, which would enable a comparison with the reference map. Signature indices and split-sample tests are

other classic methods that can be used for evaluation. Our approach is therefore entirely suitable for a calibration and evaluation of SWAT+ models.

We believe that our calibration approach will particularly improve the robustness of model calibrations if these models are to be used for the projection of catchment responses under changing environmental conditions. Botero-Acosta et al. (2022), for example, used the SWAT+ model to investigate the effects of climate change on a catchment, but had to attribute a certain degree of uncertainty to the results as there was a certain degree of equifinality regarding the calibration of the model parameters. The application of our calibration approach would be useful here in order to reduce uncertainties in the model calibration and to guarantee a physically correct behaviour of the model. This would reduce the uncertainty in the model results.

The calibration approach can also be applied to catchments with different characteristics. For catchments that are not rainfall but snowmelt dominated, the DFI method could be adapted. The calibration would then be done for the parts of smaller block lengths where the snow-fed parts of the discharge would be located. This is recommended for those catchments, as the incorporation of snowmelt is crucial for the correct discharge prediction under these circumstances (Myers et al., 2021).

Including tracer data as an additional evaluation criterion could enhance the robustness of our model parameterisation assessments (e.g. Wu et al., 2023). It offers valuable insights into discharge composition by distinguishing contributions from individual runoff components at the gauging station. For glacial and snow influenced catchments the isotope approach of Penna et al. (2014) could be applied. For wetlands, Birkigt et al. (2018) and Schwerdtfeger et al. (2016) demonstrated approaches of tracer-based modelling. This could further improve the accuracy of selecting the correct model parameterisation by including this additional evaluation step.

## 5 Conclusions

Our study shows that with our approach to calibration, a process-behavioural model parameterisation can be selected that can correctly predict the runoff and correctly map the underlying runoff-forming processes. The different performance of the various PTFs was particularly evident. These lead to widely varying results for both the runoff and the processes themselves. As part of our approach, however, it was possible to detect and sort out the PTFs that led to process depictions that did not correspond to the expected process behaviour in the catchment. This emphasises the importance for modellers to consider the use of different PTFs/soil parameterisations and a critical evaluation of those.

Our method helps to create process-behavioural models that achieve the right results for the right reasons (Beven, 2018). It improves the robustness of the model, as the model's process-behaviour can be approximated much more closely to the actual observed process-behaviour of the catchment. This could be particularly relevant if the models are to be used for the evaluation of changing environmental parameters. These include, for example, changes in land use, such as the conversion of forest into arable land, but also changes in the temperature and precipitation regime, as is the case with climate change. Our work thus contributes to the development of reliable models for the projection of catchment behaviour under future changes. However, future work is necessary to analyse to what extent better process depiction can positively influence the model prediction under external changes.

**Appendix A: Figures**

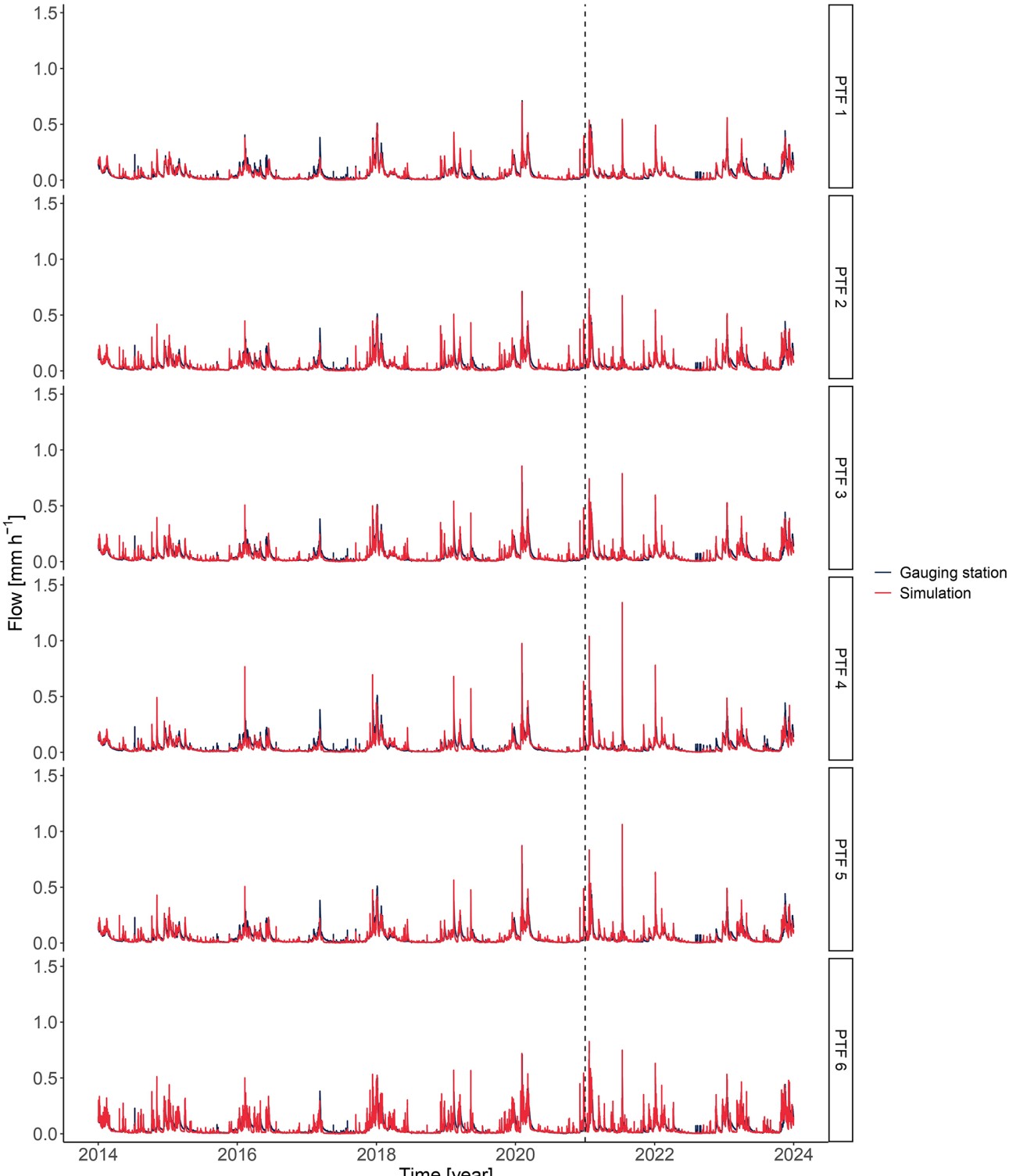

**Figure A1.** Full hydrographs for the gauging station and the simulation for PTFs 1–6. The hydrograph left of the dashed line was used as calibration period, while the part right of the dashed line served as calibration period.

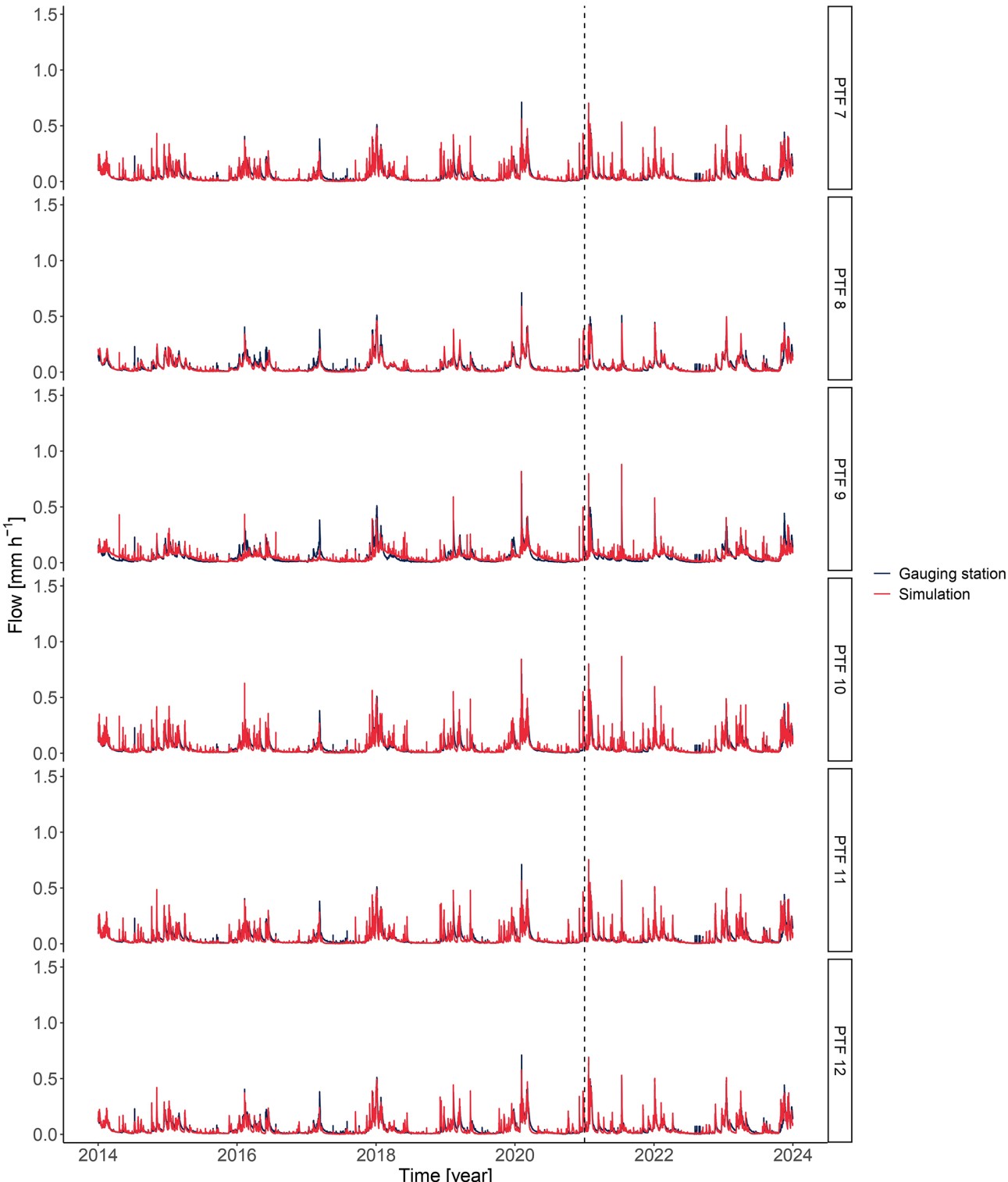

**Figure A2.** Full hydrographs for the gauging station and the simulation for PTFs 7–12. The hydrograph left of the dashed line was used as calibration period, while the part right of the dashed line served as calibration period.

## Appendix B: Tables

**Table B1.** Parameters adjusted within our parameterisation and calibration approach.

| Parameter | Unit | Values | Description |
|---|---|---|---|
| $k_x$ | $[\mathrm{m\,s^{-1}}]$ | $3 \times 10^{-5}$ | Lateral conductivity of the aquifer in $x$ direction |
| $k_y$ | $[\mathrm{m\,s^{-1}}]$ | $3 \times 10^{-5}$ | Lateral conductivity of the aquifer in $y$ direction |
| Colmation | $[\mathrm{m\,s^{-1}}]$ | $1 \times 10^{-5}$ | Hydraulic conductivity resistance between aquifer and waterbody |
| River network threshold | [–] | 50 | Threshold for the river network generation in TANALYS |
| $d_r$ | [–] | 6–75 (160) | Scaling factor for the interflow |
| $k_d$ | [h] | 2–20 | Recession parameter for the direct runoff SLR |
| $k_{ifl}$ | [h] | 5–36 | Recession parameter for the interflow SLR |

**Table B2.** Calibrated parameters with values for different PTFs.

| PTF | $k_d$ | $k_{ifl}$ | $d_r$ | Comment |
|---|---|---|---|---|
| 1 | 4 | 15 | 30 | |
| 2 | 2 | 21 | 60 | |
| 3 | 2 | 9 | 24 | |
| 4 | 20 | 23 | 6 | |
| 5 | 2 | 8 | 45 | |
| 6 | 2 | 5 | 23 | |
| 7 | 4 | 30 | 34 | |
| 8 | 2 | 36 | 50 | |
| 9 | 2 | 3 | (160+) | Calibration of $d_r$ not possible |
| 10 | 2 | 3 | (160+) | Calibration of $d_r$ not possible |
| 11 | 7 | 32 | 29 | |
| 12 | 2 | 34 | 55 | |

*Code and data availability.* The calibrated model as well as the used input data can be found under https://doi.org/10.5281/zenodo.14841047 (Heuer, 2025).

*Author contributions.* MCC and MMH conceptualised the study and methods. MMH did the data curation, formal analysis, software development, and the original draft. MCC did the funding acquisition, project administration, and supervision. MMH, HM, and MCC did the review and editing.

*Competing interests.* The contact author has declared that none of the authors has any competing interests.

ther geographical representation in this paper. While Copernicus Publications makes every effort to include appropriate place names, the final responsibility lies with the authors.

*Acknowledgements.* We thank the Stadtwerke Trier (SWT) for providing gauging data for the catchment. We also thank the Landesamt für Umwelt (LfU) Mainz for providing high-resolution climate data. We also thank Jörg Schulla for his constant support on the WaSiM model's usage.

*Financial support.* Funded by the Deutsche Forschungsgemeinschaft (DFG, German Research Foundation) project no. 426111700 and Forstliche Forschungsförderung no. 5.2-04-2023 project "Klimawald2100 Modul Wald und Wasser". The publication was funded/supported by the Open Access Fund of Universität Trier.

*Review statement.* This paper was edited by Wouter Buytaert and reviewed by Dan Myers and one anonymous referee.

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
