# Peer review of "Finding process-behavioural parameterisations of a hydrological model using a multi-step process-based calibration and evaluation scheme"

_EGUsphere, 2025_

## Author Comment (AC1)

**Author comments on RC1 - egusphere-2025-636**

MORITZ M. HEUER, HADYSA MOHAJERANI, MARKUS C. CASPER

**This study performed a multivariable calibration on a hydrologic model, that went beyond the traditionally calibrated discharge to include other components such as ET and groundwater flow. I think the authors did a good job incorporating my previous comments about bringing in more existing literature and explaining the "why" of the study. I really like the improvements the authors made. With these improvements, I believe there is now work to do to highlight and clarify the advancements of this study beyond the previous hydrology and multivariable calibration studies mentioned. I also feel that the results should go beyond the performance of the calibrated model to testing a hypothesis or comparing with other calibration approaches, to help support the statements made in the conclusion. I provide suggestions to augment the paper in this regard.**

We would like to thank the reviewer again for his important and extensive comments! We agree that the unique strengths and advancements our approach entails in comparison to others should be described in more detail. We also think that the addition of a hypothesis regarding the PTFs together with an additional evaluation enhances the value of the study.

**Page 4, line 80. I think you need to elaborate more on what makes the calibration a new approach. You mention earlier in the introduction other modeling studies that have incorporated ET, groundwater, and soils. Is the novelty here just that you used these in a new combination, or is there something more specific you can describe, such as with the use of PTF's? You also mention the "novel approach" on line 94. I think clarifying the specific novelty in more detail would help show how this goes beyond previous work.**

We agree with the reviewer that the novelty and uniqueness of our approach could be stated more clearly. We rewrote the paragraph and added explanations regarding the novel inclusion of the PTFs for soil parameterisation and their evaluation within our approach. This is senseful as the different soil parameterisations take an important part within our calibration approach, while also leading to substantially different model parameterisations. This shows how important the evaluation of the different soil parameterisations is. We hope that these aspects are better represented now.

> To address the above-mentioned challenges, our research introduces a new approach for the parameterisation and calibration of water balance models. This approach comprises the calibration of evapotranspiration patterns of different land uses based on remote sensing ETa data, ensuring correct ETa patterns and a closed water balance. In addition, the ground water behaviour is assessed by deriving the long term baseflow from the measured discharge of the catchment. This allows for calibration of the groundwater behaviour (storage, recession) as well as the groundwater recharge (deep percolation) within the model. Furthermore, the influence of the soil parameterisation on the spatial pattern of runoff generation is assessed. This ensures a correct depiction of runoff patterns over the catchment area. Lastly, high discharge volume is calibrated by deriving information about the catchment discharge characteristics from the flow duration curve. These different methods are applied to model parameterisations whose soil hydrological properties are determined differently by a variety of pedotransfer functions. Therefore, the suitability of individual PTFs to correctly describe the soil properties of the catchment can be evaluated. By incorporating the calibration and evaluation of these different model aspects, we aim at reaching a model calibration that correctly simulates the discharge as well as the underlying hydrological processes. This represents an advantage over black-box calibration approaches, where the calibration is not aimed at the correct representation of hydrological processes. It also extends existing multi-variable calibration concepts, which previously did not take different soil parameterisations into account in their calibration and evaluation schemes.

**Page 4, lines 91-95. I suggest adding a scientific research question and hypothesis, again to help make it clearer what the advancement of the study is, and make the novelties more distinguishable. This would help clarify what is tested in the study and what the focus is beyond the methods. For instance, what is "more accurate" in comparison to? Is there any hypothesis you could develop and test with the PTF's in the calibration?**

We thank the reviewer for the idea of adding a research question and a corresponding hypothesis. We aggregated our existing tree hypotheses into one single hypothesis (ii), and added two hypotheses that are directly adressing methodological applications within the paper (i + iii). This should enable an evaluation of the applicability and advantages of our calibration approach as well as highlighting the novelties.

> The aim of our paper is to investigate whether a multi-variable calibration approach can be used to select a model parameterisation that correctly represents both the simulated runoff and the underlying hydrological processes. We hypothesise that (i) the aquifer calibration can be derived from the measured baseflow; (ii) a model parameter set can be found that leads to correct discharge and process depiction; (iii) and soil parameterisations derived by different PTFs that lead to incorrect process depictions in the model can be reliably detected and filtered out.

**Page 24, line 452. At the end of the results, I can't help feeling that this section is cut off too soon. I'm not convinced that stopping at the model performance evaluations is sufficient. This could go back to my comment about having a hypothesis to test, to go beyond just describing a method. As an idea, would it be possible to add a comparison with a model calibrated with a traditional, discharge-only approach, or with other multivariate approaches from the literature? In terms of performance, parameterizations, land use and climate change simulation, etc. This could help provide quantitative evidence showing why a modeler should use your approach rather than other common approaches, and what the improvement could be for their work if they adopt it. Or, as another idea, clarifying and testing a hypothesis with PTF's could help accomplish this. I think it would be beneficial to update the abstract to include a sentence of the findings beyond performance evaluations (lines 16-19) as well.**

We agree with the reviewer that an additional evaluation of the method is helpful in order to demonstrate its advantages compared to traditional practices. The idea of comparing a model calibration based on our entensive approach with a calibration based on other non-extensive calibration approaches is indeed a good idea. We are currently working on this topic. However, we noticed quite early that this idea comprises a lot of work, which is why we believe it to be more suited for an additional, separate paper in the future.

However, as we added a hypothesis regarding the evaluation of PTFs, we could add an analysis of how the different PTFs perform within the different calibration steps. By this, we can demonstrate the importance of our multi-criteria approach regarding the incorporation and evaluation of the soil parameterisations derived by different PTFs.

Table 1: Evaluation of the model based on different PTF parameterisations for the three main calibration steps. A mark indicates satisfactory results for the respective step. A mark for overall benchmark is granted if all three calibration steps are marked as satisfactory.

| PTF | Groundwater model calibration | Runoff process patterns | Discharge prediction | Overall benchmark |
|---|---|---|---|---|
| 1 | X | X | X | X |
| 2 | X | X | - | - |
| 3 | X | X | - | - |
| 4 | X | - | - | - |
| 5 | X | X | - | - |
| 6 | X | - | - | - |
| 7 | X | X | - | - |
| 8 | X | X | X | X |
| 9 | - | - | - | - |
| 10 | - | - | - | - |
| 11 | X | - | - | - |
| 12 | X | X | - | - |

We added a short paragraph to the method section, explaining which requirements must be met for each step in order to be considered as satisfactory.

**2.11 Evaluation of PTF suitability**

For the evaluation of the different PTFs, the respective model performance for each calibration step was evaluated. In order to be considered as satisfactory regarding the groundwater model calibration, the PTF must allow the model's CDC to be adjusted in slope and height to match the reference curve of the gauging station. If the slope or the height can not be brought into concordance with the reference, the PTF is considered as unsatisfactory. For the evaluation of the dominant runoff process patterns, the respective $SPAEF_{mod}$ values were used as the discriminatory statistic. Here, all PTFs that lead to $SPAEF_{mod}$ values above 0.5 were considered as satisfactory. The threshold was chosen as values above 0.5 usually lead to already well fitting patterns (MOHAJERANI et al., 2023). For the evaluation of the discharge prediction, the R² NSE and KGE of the validation period are used as the discriminatory variable. PTFs are considered satisfactory when the R² value, the NSE and the KGE are above 0.7. Other studies already consider values of 0.5 as satisfactory (MORIASI et al., 2015; ROGELIS et al., 2016). However, we aimed for a model that shows stronger congruence with the reference discharge curve, therefore chosing a higher threshold value. Then, an overall benchmark was deducted based on the three individual evaluation results. Only if a PTF leas to satisfactory results over all three evaluation steps, the whole PTF was considered as satisfactory.

We also added a sentence to the abstract that elaborates on the evaluation of the different PTFs over all calibration steps.

[...] Simultaneously, our calibration approach ensured a correct depiction of the underlying processes (groundwater behaviour, runoff patterns). We were also able to detect the model parameterisations based on the PTFs that showed satisfactory results across all calibration steps. This enables a targeted selection of the most suitable PTF for determining the soil properties. This means that our calibration approach allows selecting a process-behaviourally faithful one from many possible parameterisation variants.

**Page 28, lines 531-534. I like this statement about the importance of giving attention to PTF's and how this is the first study to use them in calibration. I think this should be elaborated on earlier, such as at the end of the introduction, and could be the basis for a novel hypothesis about this narrower focus.**

We agree that the relevance of the PTF choice could be elaborated on earlier. We expanded the paragraph about PTFs in the introduction accordingly.

Pedotransfer functions (PTF) allow the estimation of soil hydraulic properties from widely available soil data like grain size, density, or depth. Simulation outcomes of different PTFs highly differ in runoff components (surface runoff, interflow and deep percolation) and evapotranspiration (ETa) rates in space and time (Refsgaard, 2001; Stisen et al., 2008; Koch et al., 2016, Koch et al., 2017; Casper et al., 2019; Mohajerani et al., 2021). Therefore, the correct choice of a PTF for soil parameterisation is crucial. Despite the knowledge about the difference the PTF choice makes, modellers seem to give this too little attention. Often, established PTFs are chosen without evaluating if they are really suitable for the soil parameterisation of the specific catchment's soils.

**Page 30, line 580 to page 31, 594. I think the conclusions section still has a way to go. The first paragraph is essentially summarizing the previous literature of the introduction. I suggest rewriting it to be more about the learnings from this study beyond that. The second paragraph discusses reasons the new approach is better than previous approaches. However, this is largely conceptual, as the new approach has not been compared with previous approaches in the results. I think adding a quantitative comparison with a different calibration approach or something similar to the results as I suggest could help provide support for these statements.**

We agree with the reviewer that the conclusion could be more specific about the findings of our paper.

Our study shows that with our approach to calibration, a process-behavioural model parameterisation can be selected that can both correctly predict the runoff and correctly map the underlying runoff-forming processes. The different performance of the various PTFs was particularly evident. These lead to widely varying results for both the runoff and the processes themselves. As part of our approach, however, it was possible to recognise and sort out the PTFs that led to process depictions that did not correspond to the expected ones in the model area. This emphasises the importance for modellers to consider the use of different PTFs / soil parameterisations and a critical evaluation of those.

Our demonstrated method helps to create process-behavioural models that achieve the right results for the right reasons (BEVEN, 2018). It improves the robustness of the model, as the model's process-behaviour can be approximated much more closely to the actual process-behaviour of the catchment. This is particularly relevant if the models are to be used for the evaluation of changing environmental parameters. These include, for example, changes in land use, such as the conversion of forest into arable land, but also changes in the temperature and precipitation regime, as is the case with climate change. Our work thus contributes to the development of reliable models for the projection of catchment behaviour under future changes.

**Title. If I'm understanding right, the current title describes doing something that other studies have done in the literature review of the introduction? (multi-variable process-based calibration of a behavioural hydrological model). If so, I suggest making the title more specific to what this study does that goes beyond the previous literature, similar to my comments above. For instance, it could be an answer to a research question or hypothesis that is tested, such as with PTF's.**

We agree that the title can be more specific for the study. We put the focus on finding the process-behavioural parameterisations and applying the evaluation scheme (for the different PTFs). While we do evaluated different PTFs in the text, we would refrain from adding this to the title, in order to keep the title concise.

Finding process-behavioural parameterisations of a hydrological model using a multi-step process-based calibration and evaluation scheme

**Page 6, line 124. How does the model handle snowmelt? It may be worth a brief explanation if snowmelt is relevant to the study period and parameters, since it's later mentioned in lines 280 and 570.**

We added an explanatory sentence to the model description. We used a relatively simple approach for our catchment. However, for catchments that are highly snowmelt or even glaciermelt-dominated, a various number of more complex snowmelt models are included within the WaSiM-model.

[...] At the bottom, a section of aquifer layers is included. Surface runoff, interflow and groundwater-contributing deep percolation can be generated. Surface runoff and interflow of each subcatchment are delayed through a single linear reservoir (SLR) each. Snowmelt is handled with a temperature-index-approach where the snowmelt rate is determined by the temperature and a melt factor.

**Table 1. Would it be possible to add another column to this table that has a brief description of the PTF's for each row? For instance, something to help explain why the dominant runoff process patterns (interflow, surface runoff, deep percolation, etc.) could be different among the PTF's in section 3.3.**

We agree that such an overview would be helpful. However, there isn't a single variable or characteristic that is the basis or reason for why the different PTFs behave so differently. This is a result of the different dataset sizes, the specific used soils for training the PTFs, then there are some PTFs that are based on regular regression models while others are neural network based. This is why we refer to MOHAJERANI et al. (2021) for a more detailed analysis of the PTFs that we used. We expanded the corresponding text passage in the method section to increase the clarity about this.

[. . .] For PTF combinations 8 to 12, the values were calculated by the respective PTF's equation for $k_{sat}$. The chosen PTFs mainly differ in their underlying soil data, sample size, and considered soil parameters for the resulting predictive equations. Also, some PTFs are based on regular regression models while others are based on neural networks for deriving the hydraulic parameter values. A comprehensive analysis of the characteristics of PTFs 1 to 11 and their impact on the derived hydrological soil properties has been provided by MOHAJERANI et al. (2021).

**Page 18, lines 350-360. As these performance metrics are commonly used, I suggest to remove these equations and descriptions, and just direct readers who are interested to the references.**

We will remove the formulas for the revised version.

**Page 29, line 540 to page 30, line 579. I got a lot out of this section. I thought it answered well the questions I had about replicability and transferability from my previous comments.**

Thank you very much! We are pleased that the changes we made based on your comments answer your questions and increased clarity!

**References**

Beven, K. J. (2018). On hypothesis testing in hydrology: Why falsification of models is still a really good idea. *Wiley Interdisciplinary Reviews: Water*, *5*(3), e1278.

Mohajerani, H., Jackel, M., Salm, Z., Schütz, T., & Casper, M. C. (2023). Spatial Evaluation of a Hydrological Model on Dominant Runoff Generation Processes Using Soil Hydrologic Maps. *Hydrology*, *10*(3), 55.

Mohajerani, H., Teschemacher, S., & Casper, M. C. (2021). A comparative investigation of various pedotransfer functions and their impact on hydrological simulations. *Water*, *13*(10), 1401.

Moriasi, D. N., Gitau, M. W., Pai, N., & Daggupati, P. (2015). Hydrologic and water quality models: Performance measures and evaluation criteria. *Transactions of the ASABE*, *58*(6), 1763–1785.

Rogelis, M. C., Werner, M., Obregón, N., & Wright, N. (2016). Hydrological model assessment for flood early warning in a tropical high mountain basin. *Hydrology and Earth system sciences discussions*, *2016*, 1–36.

---

## Author Comment (AC2)

**Author comments on RC2 - egusphere-2025-636**

MORITZ M. HEUER, HADYSA MOHAJERANI, MARKUS C. CASPER

**In this manuscript, Heuer et al. present a multi-step calibration approach and test 12 parametrizations of pedotransfer function for plausibility with regard to spatial patterns of simulated process dominance.**

**I like the approach of the authors and think that it is a valuable contribution to process-based model calibration. I have only a few comments.**

We would like to thank the reviewer for their helpful and valuable comments! We are glad the reviewer likes our approach and believes it to be a valuable contribution to process-based model calibration. We are certain the proposed changes improve the manuscript's clarity.

**It is not always clear whether the authors are analysing evapotranspiration (ETa) or evaporation. It is important to clarify this and to make clear that the comparison between MODIS data and simulation results is valid. See for example L. 10 "Evapotranspiration parameters based on MODIS evaporation data". See also L. 178-180.**

Thank you for pointing out these ambiguations. The MODIS data we refer to are evapotranspiration time-series and not just evaporation measurements. We will change the mentioned text lines within the manuscript to evapotranspiration in order to increase clarity.

**The selection of the hydrograph efficiency metrics is not convincing. NSE and R2 are conceptually very similar. There is no advantage in using R2 in addition to NSE. Table 5 shows that a lower NSE results in a lower R2 and vice versa. So, R2 is not needed. I suggest adding a complementary metric e.g. PBIAS to capture another part of the hydrological cycle. Another option could be to use the three components of the KGE separately (as KGE-beta represents the same hydrological behaviour as PBIAS).**

We thank the reviewer for this proposal. We will replace the $R^2$ metric with the $PBIAS$ to include a complementary metric to the $NSE$ and $KGE$ metrics. This results in the following reworked efficiency metrics table:

| PTF | MHd | HDHO | $NSE_{cal}$ | $KGE_{cal}$ | $PBIAS_{cal}$ | $NSE_{val}$ | $KGE_{val}$ | $PBIAS_{val}$ |
|---|---|---|---|---|---|---|---|---|
| 1 | 3.19 | 0.9 | 0.86 | 0.89 | -5.31 | 0.78 | 0.86 | 2.64 |
| 2 | 4.88 | 0.7 | 0.66 | 0.82 | -3.79 | 0.59 | 0.79 | 1.28 |
| 3 | 3.59 | 0.8 | 0.72 | 0.84 | -6.56 | 0.63 | 0.81 | -0.15 |
| 4 | 3.36 | 0.5 | 0.67 | 0.79 | -9.30 | 0.45 | 0.69 | -5.25 |
| 5 | 3.68 | 0.7 | 0.72 | 0.83 | -8.37 | 0.55 | 0.75 | -3.16 |
| 6 | 6.60 | 0.6 | 0.49 | 0.73 | -4.46 | 0.34 | 0.70 | 5.01 |
| 7 | 5.04 | 0.8 | 0.60 | 0.81 | -0.04 | 0.49 | 0.77 | 8.67 |
| 8 | 2.76 | 0.8 | 0.87 | 0.89 | -5.52 | 0.78 | 0.87 | 3.16 |
| 9 | 6.68 | 0.8 | 0.54 | 0.61 | -5.38 | 0.49 | 0.60 | -3.24 |
| 10 | 2.01 | 0.6 | 0.59 | 0.78 | -4.94 | 0.47 | 0.75 | 3.16 |
| 11 | 5.36 | 0.8 | 0.55 | 0.76 | -3.65 | 0.43 | 0.74 | 5.39 |
| 12 | 5.84 | 0.9 | 0.70 | 0.83 | -5.00 | 0.64 | 0.80 | 0.64 |

Together with the changes in the table, we change the methodology and result parts where the used efficiency metrics are described and evaluated.

We changed lines 349 to 351 to:

> The third metric included was the PBIAS. This metric is a measurement of the average tendency of the simulated data to be larger or smaller than their observed counterparts (GUPTA et al., 1999).

We changed lines 428 to 433 to:

> The model shows high values for the efficiency measures for both the calibration and the validation period. Between calibration and validation, there is only a slight decrease in the NSE from 0.87 to 0.78, while the KGE decreases only minimally from 0.89 to 0.87. Values for the PBIAS slightly improve from -5.52% for the calibration period to 3.16% for the validation period. Efficiency measures for the split-sample test of other PTFs (Table 5) show a large value range. For example, PTF 1 also shows relatively high values for the NSE and KGE. However, PTFs 4, 6, 9, 10 and 11 show low values. All other PTFs show values in between. The PBIAS shows values of around -5.00% for most PTFs for the calibration period, while the values for the validation period are between -5 and 5% for all PTFs except PTF 7.

We changed lines 516 to 517 to:

> This is also supported by high values of efficiency measures such as NSE (0.78), KGE (0.87) and a low PBIAS (3.16%) for the validation period in the split-sample test.

**L.371-374: This part needs to be reformulated. Three sentences mention that PTF 9 and 10 are not valid. I suggest to make one clear statement.**

> [...] This was achieved by using a single layer aquifer with a thickness of 1 m and lateral hydraulic conductivities of $3E - 5 \cdot m \cdot s^{-1}$. In step 3, the CDC height could also be adapted to the course of the gauging station curve for almost all PTFs except PTF 9 and 10. The corresponding calibrated values for $d_r$ range from 6 for PTF 4 up to 60 for PTF 2. In the front part of the curve, the simulations almost exclusively run below the reference curve of the gauging station.

**Table 5: Use consistently two digit numbers. For PTF1 and KGEval three digit numbers are shown.**

We thank the reviewer for pointing this out! We corrected the values' decimal points to be the same for all entries.

**References**

Gupta, H. V., Sorooshian, S., & Yapo, P. O. (1999). Status of automatic calibration for hydrologic models: Comparison with multilevel expert calibration. *Journal of hydrologic engineering*, *4*(2), 135–143.